# Spontaneous neuronal oscillations in the human insula are hierarchically organized traveling waves

**Anup Das[1]\*[†], John Myers[2]\*[†], Raissa Mathura[2], Ben Shofty[2], Brian A Metzger[2], Kelly Bijanki[2], Chengyuan Wu[3], Joshua Jacobs[1,4]\*[‡], Sameer A Sheth[2]\*[‡]**

[1]Department of Biomedical Engineering, Columbia University, New York, United States; [2]Department of Neurosurgery, Baylor College of Medicine, Houston, United States; [3]Department of Neurosurgery, Thomas Jefferson University, Philadelphia, United States; [4]Department of Neurological Surgery, Columbia University, New York, United States

**\*For correspondence:**
ad3772@columbia.edu (AD);
John.Myers@bcm.edu (JM);
joshua.jacobs@columbia.edu
(JJ);
Sameer.Sheth@bcm.edu (SAS)

[†]These authors contributed
equally to this work
[‡]Joint senior authors

**Competing interest:** The authors
declare that no competing
interests exist.

**Reviewing Editor:** Shella
Keilholz, Emory University and
Georgia Institute of Technology,
United States

**Abstract** The insula plays a fundamental role in a wide range of adaptive human behaviors, but its electrophysiological dynamics are poorly understood. Here, we used human intracranial electroencephalographic recordings to investigate the electrophysiological properties and hierarchical organization of spontaneous neuronal oscillations within the insula. We analyzed the neuronal oscillations of the insula directly and found that rhythms in the theta and beta frequency oscillations are widespread and spontaneously present. These oscillations are largely organized along the anterior–posterior (AP) axis of the insula. Both the left and right insula showed anterior-to-posterior decreasing gradients for the power of oscillations in the beta frequency band. The left insula also showed a posterior-to-anterior decreasing frequency gradient and an anterior-to-posterior decreasing power gradient in the theta frequency band. In addition to measuring the power of these oscillations, we also examined the phase of these signals across simultaneous recording channels and found that the insula oscillations in the theta and beta bands are traveling waves. The strength of the traveling waves in each frequency was positively correlated with the amplitude of each oscillation. However, the theta and beta traveling waves were uncoupled to each other in terms of phase and amplitude, which suggested that insular traveling waves in the theta and beta bands operate independently. Our findings provide new insights into the spatiotemporal dynamics and hierarchical organization of neuronal oscillations within the insula, which, given its rich connectivity with widespread cortical regions, indicates that oscillations and traveling waves have an important role in intrainsular and interinsular communications.

## Editor's evaluation

This manuscript is of interest to neuroscientists willing to deepen their knowledge related to the role of the insula and to any scientist interested in oscillatory activity and traveling waves in the brain. The substantial dataset and the novel methodological approach provide interesting insights on the functional organization of this brain region.

## Introduction

The insula plays a critical role in cognitive control (*Menon and Uddin, 2010*), awareness (*Craig, 2009*; *Critchley et al., 2004*), and adaptive human behaviors (*Menon and Uddin, 2010*; *Singer et al., 2009*). Dysfunction of the human insula is prominent in many psychiatric and neurological disorders

(*Jilka et al., 2014*; *King-Casas et al., 2008*; *Sha et al., 2019*). Human functional magnetic resonance imaging (fMRI) studies have demonstrated insula involvement in detection and attentional capture of goal-relevant stimuli across a wide range of cognitive tasks (*Cai et al., 2014*; *Dosenbach et al., 2008*; *Dosenbach et al., 2006*; *Menon and Uddin, 2010*; *Sridharan et al., 2008*). Despite its critical role in cognitive function and dysfunction (*Menon, 2011*; *Uddin, 2015*), the electrophysiological foundations of the insula, its hierarchical organization, and sub-second spatiotemporal dynamics are poorly understood, in part due to the low temporal resolution of fMRI data. Neuronal oscillations are a fundamental feature of cortical networks and are critical for facilitating communication among brain areas (*Berger, 1929*). Understanding the fundamental features of insula oscillations is therefore important, given its rich connectivity with widespread cortical regions (*Menon and Uddin, 2010*). Here, we address these challenges and investigate, for the first time, the hierarchical organization and spatiotemporal gradients in the insula using a unique human intracranial electroencephalography (iEEG) dataset.

Human fMRI studies have shown that the insula plays a foundational role in human cognitive processes by interacting with other large-scale brain networks such as the medial default mode network (DMN) and lateral frontal–parietal network (FPN), which are two other large-scale brain networks that play an important role in external stimulus-related cognition and monitoring internal mental processes (*Menon, 2011*; *Power et al., 2011*). Crucially, the intrinsic connectivity of the insula displays close correspondence with task-related coactivation patterns (*Sridharan et al., 2008*; *Supekar and Menon, 2012*), and this correspondence has allowed intrinsic and task-related connectivity associated with the insula to be demarcated and studied under a common, triple-network model framework (*Menon, 2011*).

Cytoarchitectonics studies in non-human primates have shown functional subdivisions within the insula. These studies revealed that the non-human primate insula can be divided into an anterior-basal agranular part and a posterior granular part (*Augustine, 1996*; *Mesulam and Mufson, 1985*). These studies, as well as direct tract-tracing studies, have also shown that the anterior insula (AI) is structurally connected to orbitofrontal, entorhinal, amygdala, periamygdaloid, piriform, olfactory cortices, and the temporal pole whereas the posterior insula (PI) is connected to primary and secondary sensory and motor cortices (*Fudge et al., 2005*; *Höistad and Barbas, 2008*; *Mesulam and Mufson, 1985*; *Stefanacci and Amaral, 2002*). Similar to the findings in non-human primates, MRI studies in humans have also shown subdivisions in the insula (*Ghaziri et al., 2017*; *Morel et al., 2013*). Previous fMRI studies have shown functional subdivisions within the insula, as the AI was involved in task-related engagement and disengagement with the DMN and FPN across a wide range of cognitive and affective tasks (*Cai et al., 2016*; *Cai et al., 2021*; *Chen et al., 2016*; *Lamm and Singer, 2010*; *Sridharan et al., 2008*). The AI, along with the anterior cingulate cortex, is known as the salience network, which is activated simultaneously across most cognitive task domains, thus forming a core task-set system (*Dosenbach et al., 2008*; *Dosenbach et al., 2006*; *Ouyang et al., 2008*). Together, these findings suggest a common hierarchical organization within the insula across species.

Neuronal oscillations are a fundamental feature of cortical networks and are critical for facilitating communication among a wide range of brain areas across species (*Berger, 1929*). Understanding the fundamental features of insula oscillations is therefore important, given its rich connectivity with widespread cortical regions (*Menon and Uddin, 2010*). However, the electrophysiological basis of the functional subdivisions in the insula and its spatiotemporal gradients and hierarchical organization of spontaneous neuronal oscillations are poorly understood. iEEG recordings with adequate sampling of electrodes from both the anterior and posterior subdivisions of the insula provide one way to address this challenge. We examined a resting-state iEEG dataset of 10 participants who underwent treatment for intractable epilepsy and had a total of 239 electrodes implanted across both the anterior and posterior subdivisions of the insula. The first goal of our study was to determine the hierarchical spatial organization of the electrophysiological patterns across the insula and, in particular, explaining how they relate to the frequency and power of neuronal oscillations. Global brain intrinsic gradients have been known to be closely associated with the structural connectivity of the brain areas and the degree of task-related activations (*Huntenburg et al., 2018*; *Paquola et al., 2019*; *Zhang et al., 2019*). Analyses of intrinsic oscillatory gradients in the insula are, therefore, crucial to understand the structure–function relationship of areas within the insula and between the insula and other regions of the cortex (*Huntenburg et al., 2018*; *Vázquez-Rodríguez et al., 2019*) and provide a more mechanistic understanding of oscillations as a means of intrainsular and interinsular communications in the human brain.

Based on recent findings that showed a brain-wide posterior-to-anterior frequency gradient and an anterior-to-posterior power gradient using resting-state magnetoencephalography (MEG) recordings from humans (*Mahjoory et al., 2020*), as well as similar findings from direct recordings at the cortical surface (*Zhang et al., 2018*), we hypothesized that the insula would have a posterior-to-anterior frequency gradient and an anterior-to-posterior power gradient as well.

Recent studies in humans suggested that the hierarchical organization of neuronal activity across the cortex is not a stationary phenomenon, rather that it has a dynamic phenomenon, which propagates across the cortex in space and time (*Alamia and VanRullen, 2019*; *Halgren et al., 2019*; *Hangya et al., 2011*; *Kleen et al., 2021*; *Lozano-Soldevilla and VanRullen, 2019*; *Massimini et al., 2004*; *Muller et al., 2016*; *Stolk et al., 2019*; *Zhang and Jacobs, 2015*; *Zhang et al., 2018*). This phenomenon, also known as 'traveling waves,' consists of oscillations that propagate progressively across the cortex. Traveling waves have been shown to be relevant for visual perception (*Besserve et al., 2015*; *Davis et al., 2020*; *Gabriel and Eckhorn, 2003*; *Muller et al., 2014*; *Nauhaus et al., 2009*; *Townsend et al., 2015*; *Vinck et al., 2010*; *Zanos et al., 2015*) and movement initiation (*Balasubramanian et al., 2020*; *Denker et al., 2018*; *Rubino et al., 2006*; *Rule et al., 2018*; *Takahashi et al., 2015*) in non-human primates and visual processing (*Liang et al., 2021*; *Stroh et al., 2013*; *Xu et al., 2007*) and spatial navigation (*Agarwal et al., 2014*; *Patel et al., 2012*; *Patel et al., 2013*) in rodents. In humans, traveling waves have been observed during sleep (*Dickey et al., 2021*; *Hangya et al., 2011*; *Massimini et al., 2004*; *Muller et al., 2016*) as well as during behavior (*Alamia and VanRullen, 2019*; *Kleen et al., 2021*; *Stolk et al., 2019*; *Zhang and Jacobs, 2015*; *Zhang et al., 2018*). However, it is unclear whether these traveling waves, which are mostly observed on the surface of the cortex, are also relevant for deep neocortical structures such as the insula.

In light of this emerging literature, the second goal of our study was to detect traveling waves within the human insula. We adopted a novel method using circular statistics to identify traveling waves amidst the neuronal oscillations that we measured in each participant individually. In addition to detecting traveling waves, we also analyzed differences in the characteristics of the observed traveling waves across participants, and we measured traveling waves at a range of frequencies. Based on findings from human electrocorticography studies that showed low-frequency alpha and high-frequency beta traveling waves were weakly or uncorrelated with each other in the sensorimotor cortex (*Stolk et al., 2019*), we tested whether there was a link between the properties of traveling waves in the human insula between different frequencies. Our findings provide the first electrophysiological evidence for the presence of spontaneous oscillatory gradients and traveling waves in the human insula.

## Results

### Theta frequency and power gradients in the insula

Our first goal was to investigate the spatial properties of the frequency and power gradients in the neuronal oscillations in the insula. These analyses were inspired by recent findings showing a posterior-to-anterior fast-to-slow frequency gradient and an anterior-to-posterior power gradient in oscillations measured from resting-state MEG recordings from humans (*Mahjoory et al., 2020*) and a hippocampal posterior-to-anterior frequency gradient using iEEG recordings in human spatial navigation (*Goyal et al., 2020*). We examined a dataset of subjects with iEEG recordings conducted during rest and used spectral analyses to measure the peak frequency and spectral power of the oscillations from each electrode in the insula (Materials and methods). Individual subjects in this dataset had multiple electrodes in their insula, including contacts in both hemispheres (*Figure 1*). To characterize the properties of the insula oscillations in these data, for each electrode we first detected oscillations by measuring the frequency bands where power exceeded one standard deviation above the 1/f spectrum using the multiple oscillations detection algorithm (MODAL) (*Watrous et al., 2018*). Theta (~6–9 Hz) and beta (~15–32 Hz) were the most prominent oscillations in the insula (*Figure 1*). However, there were no 9–15 Hz oscillations in the insula (*Figure 1*). In the left hemisphere,~90% of electrodes (n=107) showed a peak frequency in the theta range and 99% showed a peak in the beta frequency band. In the right hemisphere,~80% of electrodes (n=95) had theta-band peaks and ~81% had beta-band peaks. We next conducted tests of these theta and beta oscillations to evaluate their spatial and hierarchical organization.

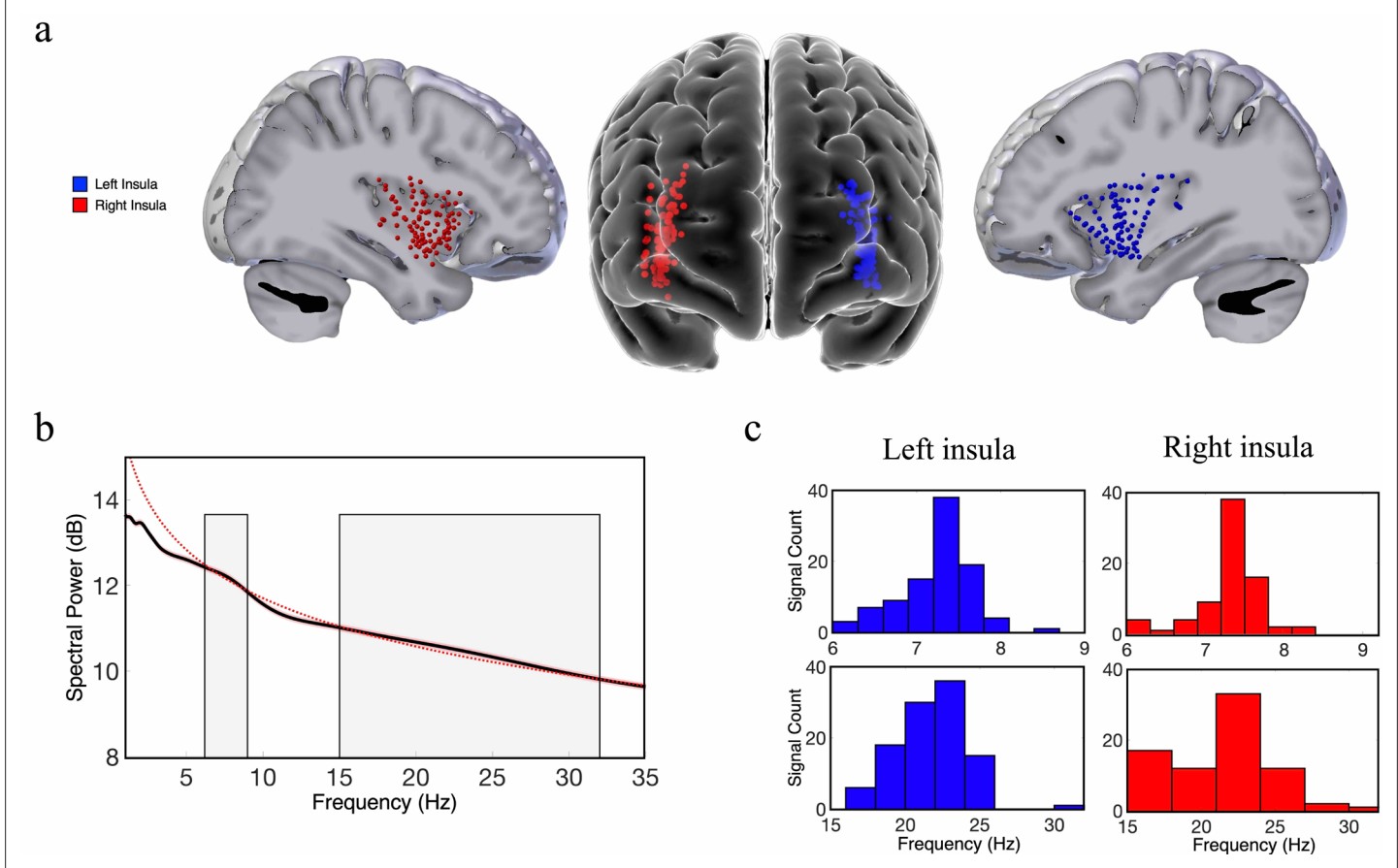

**Figure 1.** Theta-band and beta-band frequency oscillations in the human insula. (**a**) Lateral and frontal views of the brain showing electrodes in the left insula (n=125) and right insula (n=114) across all participants. (**b**) An oscillation-detection algorithm known as multiple oscillations detection algorithm (MODAL) detected theta (~6–9 Hz) and beta (~15–32 Hz) oscillations across all electrodes in the insula (Materials and methods). The black line depicts the power spectrum, the red dotted line depicts the 1/f regression line, and the red shaded region around the spectrum represents the standard error of the mean power for each frequency. Gray shaded regions indicate the detected frequency bands at which the power spectrum was higher than the 1/f background signal. (**c**) The distribution of the peak frequencies detected across all insula electrodes. In the left insula, 90% of electrodes had theta oscillations and 99% had beta oscillations. In the right insula, 80% of electrodes had theta oscillations and 81% had beta oscillations. Peak frequencies of theta and beta oscillations appear to have relatively normal distributions across the insula.

The online version of this article includes the following figure supplement(s) for figure 1:

**Figure supplement 1.** Histograms of *t*-statistics across subjects.

**Figure supplement 2.** Proportion of contacts in each anatomical segment of the insula.

To localize the electrode positions within the insula while accounting for inter-subject anatomical variability (*Kurth et al., 2010*; *Naidich et al., 2004*), we used a novel approach of modeling the shape of each subject's insula and then transforming each subject's electrode position to a standard Montreal Neurological Institute (MNI) coordinate space using an iterative, semiautomated coregistration approach (Materials and methods). Using this pooled dataset, we then evaluated how the spectral dynamics (frequency and power) of the measured neuronal oscillations correlated with the coordinates of the electrodes within the insula (y-axis and z-axis) across all participants, using linear mixed-effect (LME) models (see Materials and methods). The y-coordinate of each electrode represents its position along the AP axis and the z-coordinate corresponds to the superior–inferior (SI) axis. We separately measured oscillations within the left and right hemispheres because interhemispheric differences in structure and function could affect oscillatory organization (*Jakab et al., 2012*; *Sridharan et al., 2008*).

The AP axis coordinates (y-coordinates) of electrodes within the left insula predicted the peak frequency and power of the measured oscillations in the theta band (*Figure 2*). In the left insula, the

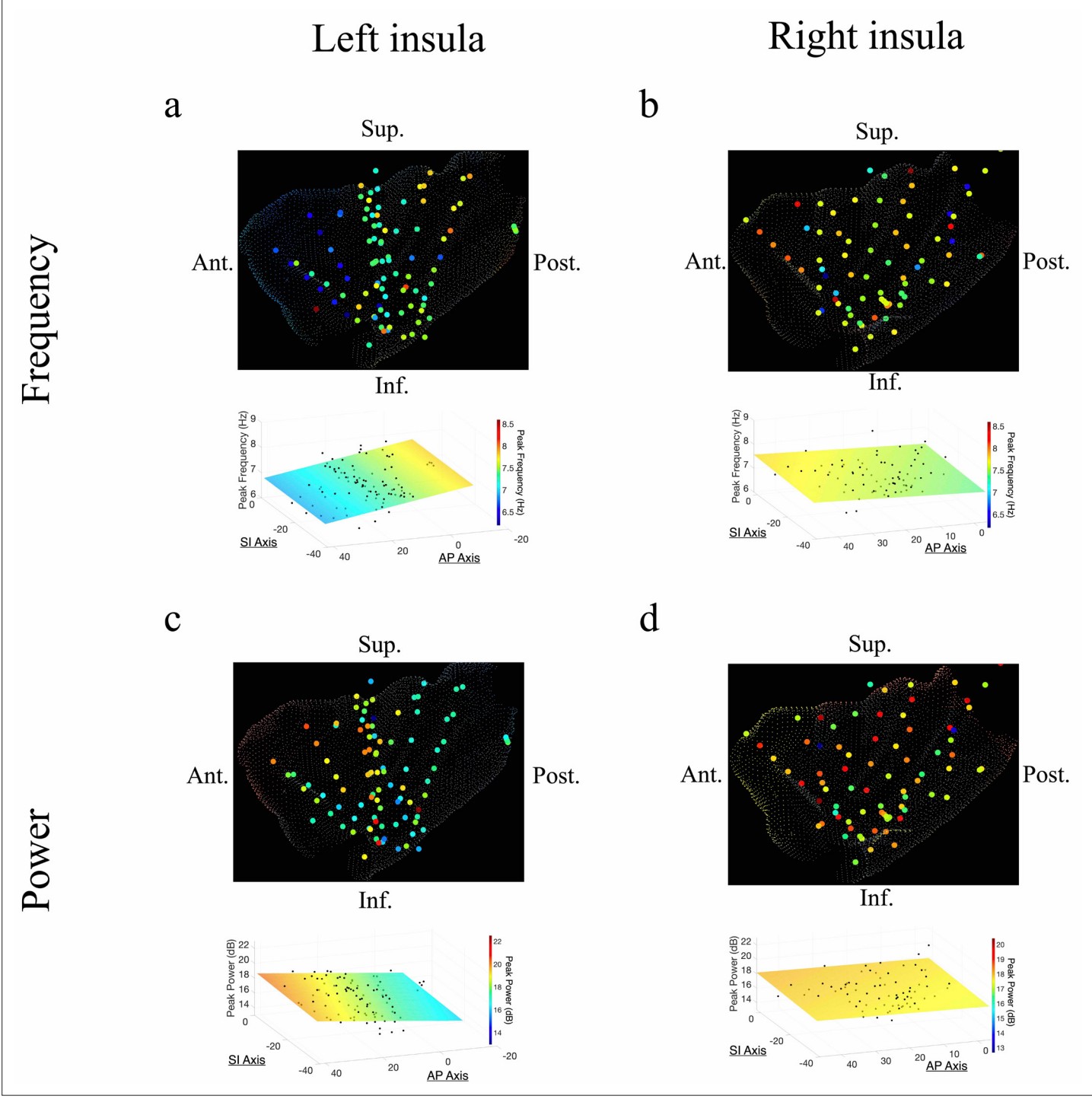

**Figure 2.** Frequency and power gradients of theta oscillations in the insula. (**a**) Peak frequencies within the theta band (~6–9 Hz) showed posterior-to-anterior gradient in the left insula. Coordinates of electrodes along the AP axis predicted peak theta frequency, where slower theta frequencies were more prominent in the anterior insula and faster theta frequencies were prominent posteriorly ($t[74]=-4.09$, $p<0.001$). (**b**) Theta frequencies in the right insula were not hierarchically organized along the AP axis ($t[57]=-0.14$, $p>0.05$), or the superior–inferior (SI) axis ($t[57]=0.07$, $p>0.05$). (**c**) Theta power showed anterior-to-posterior gradient in the left insula with theta power being higher in the anterior insula ($t[74]=2.52$, $p<0.05$). (**d**) Theta power showed a superior-to-inferior gradient in the right insula where higher amplitude theta oscillations were prominent in the superior portion of the insula ($t[57]=2.92$, $p<0.01$). A marginally significant AP gradient suggests that theta power was slightly more concentrated in the right anterior insula, similar to the power gradient in the left hemisphere ($t[57]=1.98$, $p=0.053$). No interaction effect was observed between AP and SI axes ($t[57]=-1.14$, $p>0.05$). Results were computed using linear regression where the y-axis (AP) and z-axis (SI) electrode coordinates were used as continuous independent variables to predict spectral power and frequency dynamics. Variability across participants has been accounted for in all *t*-statistics. Images were generated via automatic selection of insular vertices from anatomical parcellations using the Desikan–Killiany (DK) atlas.

frequency of theta was slower in the anterior insula ($t$[74] = –4.09, p<0.001, LME model) (*Figure 2a*); the amplitude of theta oscillations was higher in the left anterior insula ($t$[84] = 2.52, p<0.05) (*Figure 2c*). There was no indication of theta frequency ($t$[74] = –0.56, p>0.05) or theta power ($t$[74] = –0.19, p>0.05) gradients along the SI axis of the left insula.

In the right insula, the frequencies of theta oscillations were not hierarchically organized along the AP axis ($t$[57]=–0.14, p>0.05) or the SI axis ($t$[57]=0.07, p>0.05) (*Figure 2b*). However, theta power trended toward an AP gradient, where theta power was greater in the right anterior insula ($t$[57]=1.98, p=0.053) (*Figure 2d*), consistent with the left hemisphere. Theta power also showed a significant SI gradient with higher amplitudes in the superior portion of the right insula ($t$[57]=2.92, p<0.01) (*Figure 2d*).

## Beta frequency and power gradients in the insula

In addition to theta oscillations, we also observed widespread insula oscillations in the beta band. In the left insula, the power of beta oscillations showed an anterior-to-posterior gradient, with higher power anteriorly ($t$[84]=3.07, p<0.01) (*Figure 3c*). However, the frequency of beta rhythms was not hierarchically organized along the AP axis ($t$[84]=–0.95, p>0.05) or the SI axis ($t$[84]=–0.59, p>0.05) in the left insula (*Figure 3a*). In the right insula, beta power also showed a significant anterior-to-posterior gradient ($t$[58]=2.97, p<0.01) (*Figure 3d*), with higher power oscillations in the anterior insula. Thus, beta amplitude was greater in the anterior insula across both hemispheres. However, in the right hemisphere, the frequency of insula beta rhythms showed no gradients, in neither the AP ($t$[58]=–0.63, p>0.05) nor SI directions ($t$[58]=–1.10, p>0.05) (*Figure 3b*).

These findings reveal that spontaneous theta and beta oscillations are spatially organized across the insula, especially along the AP axis. In both hemispheres, higher amplitude beta oscillations were concentrated anteriorly, whereas theta frequency and power showed opposing AP gradients in the left insula. More importantly, we used LME models to analyze these patterns, which included a categorical factor for subject identity and thus ensured that our results were not caused by inter-subject differences. Therefore, all reported $t$-statistics above specifically reflect the overall strength (effect size) and spatial direction of spectral gradients across electrodes, while accounting for differences between individual subjects. The number of electrodes contributed by each subject was also similar (mean=23.9 contacts, standard error=±2.43 contacts). Notably, the single-subject gradients also directionally align with the group level results (*Figure 1—figure supplement 1*).

## Oscillation clusters are traveling waves in the human insula

Previous work on the computational modeling of weakly coupled oscillators has shown that traveling waves can naturally emerge from spatially varying gradients of oscillations across different frequencies (*Ermentrout and Kopell, 1984*). Therefore, given that we observed spatial gradients in the amplitude and frequencies of insula oscillations, we next examined the phase dynamics of these signals to test whether the frequency gradients contributed to traveling waves. A prerequisite for identifying a neural traveling wave is that a spatially contiguous region of cortex must show an oscillation at a single frequency. Thus, as a first step to identify traveling waves in the human insula, we sought to find oscillation clusters in each participant. We defined an oscillation cluster as a spatially contiguous group of electrodes having narrowband oscillations at a given frequency band (see Materials and methods). To identify electrode clusters, as in earlier work (*Zhang et al., 2018*), we first computed the mean power spectrum of each electrode and then removed the 1/f background signal to find narrowband oscillation peaks at a particular frequency. We then used a spatial clustering algorithm to identify all the groups of contiguous electrodes that showed narrowband oscillations at the same frequency. We defined these groups of electrodes, adjacent in both frequency and space, as an oscillation cluster.

*Figure 4* shows examples of oscillation clusters in the theta and beta frequency bands for the insula electrodes in participant #3. We found 28 oscillation clusters (15 in the left insula and 13 in the right insula) across 10 participants (see *Figure 4—figure supplements 1–9* for oscillation clusters from other participants). Oscillation clusters were present at both theta and beta frequencies, with 14 oscillation clusters in the theta band (8 in the left insula and 6 in the right insula) and 14 oscillation clusters in the beta band (7 in the left insula and 7 in the right insula), in 9 out of 10 participants (participant #4 did not have any oscillation clusters). Overall, 79% of all electrodes (186 out of 239 electrodes) participated in at least one oscillation cluster.

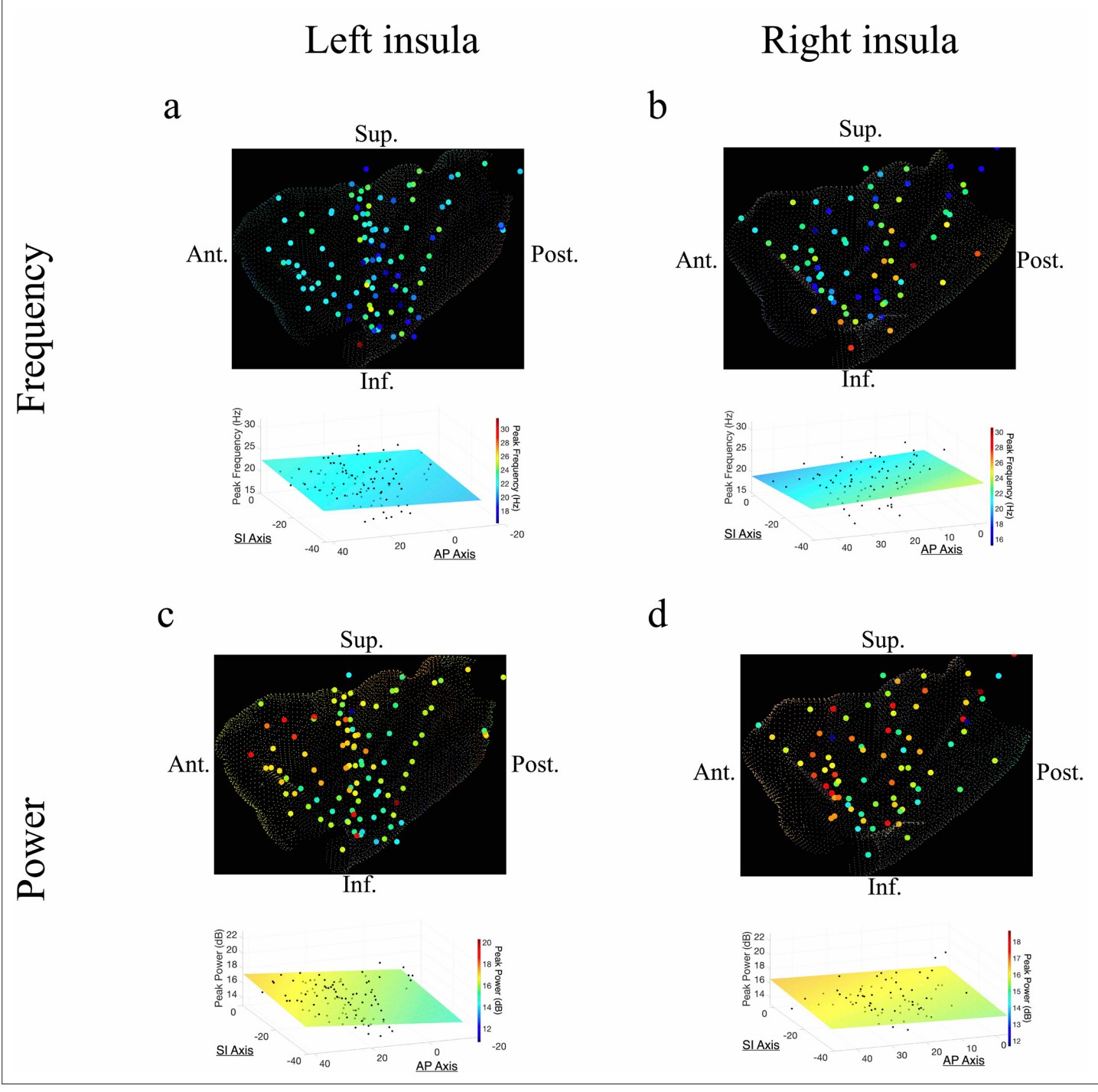

**Figure 3.** Frequency and power gradients of beta oscillations in the insula. (**a**) Peak frequencies within the beta band (~15–32 Hz) showed no spatial organization in the left insula (*t*[84]=–0.95, p>0.05). (**b**) Beta frequencies in the right insula did not show AP (*t*[58]=–0.63, p>0.05), or SI gradients (*t*[58]=–1.10, p>0.05). (**c**) Electrode coordinates along the AP axis of the left insula predicted beta power, revealing that higher power oscillations were prominent in the anterior insula (*t*[84]=3.07, p<0.01). (**d**) Beta power in the right insula also showed an AP gradient where higher amplitude beta oscillations were prominent in the right anterior portion (*t*[58]=2.97, p<0.01). Results were computed using linear regression where the y-axis (AP) and z-axis (SI) electrode coordinates were used as continuous independent variables to predict spectral power and frequency dynamics. Variability across participants has been accounted for in all *t*-statistics. Images were generated via automatic selection of insular vertices from anatomical parcellations using the Desikan–Killiany (DK) atlas.

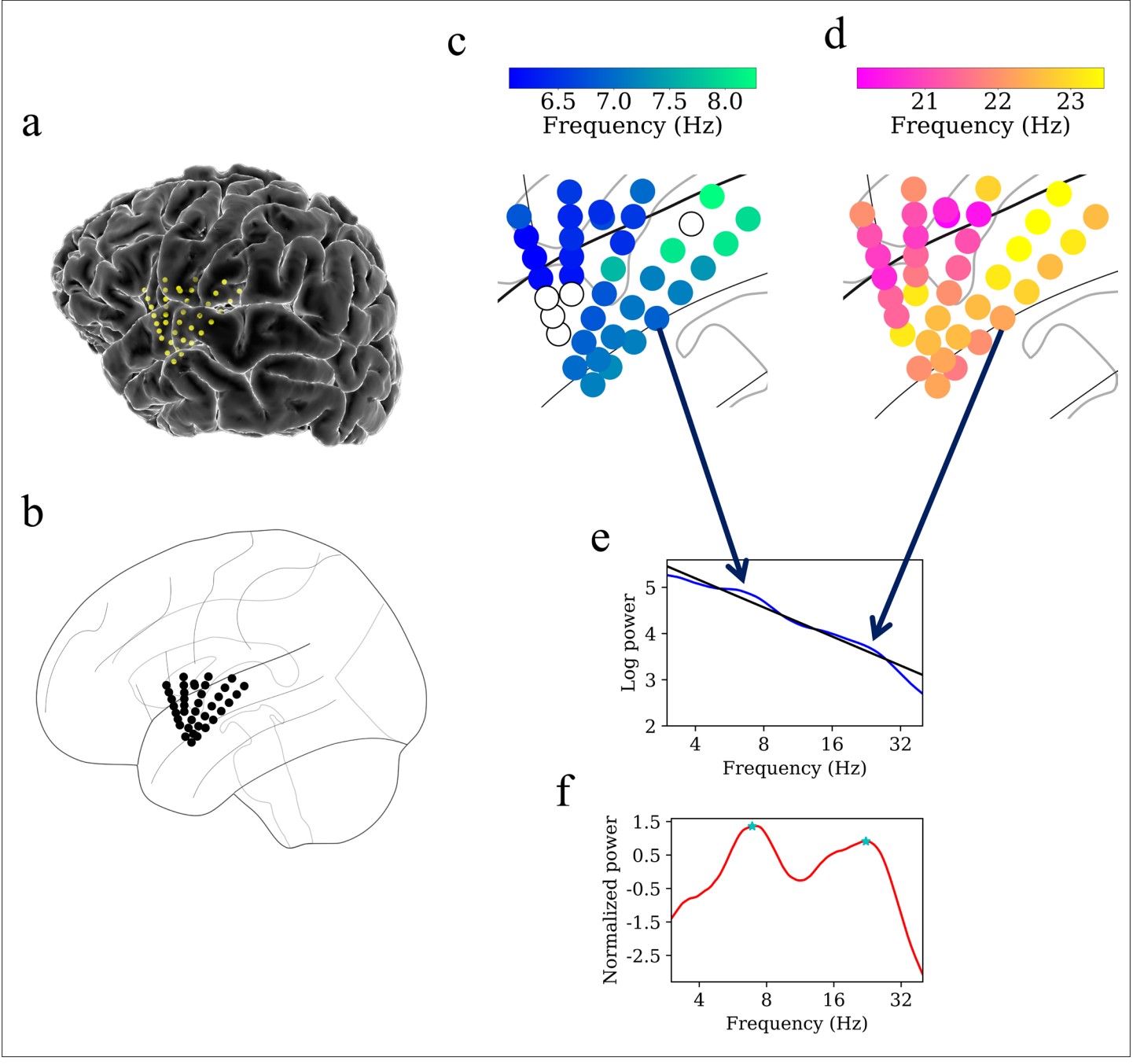

**Figure 4.** Example of spatiotemporal clustering in the human insula. A cluster was identified as a spatially contiguous groups of electrodes that had narrowband oscillations at either theta or beta frequency band (Materials and methods). (**a**) Insula electrodes visualized in a brain surface plot in participant #3. (**b**) Insula electrodes projected on to a flat brain which was subsequently used for visualization of oscillation clusters and traveling waves. (**c**) Theta frequency oscillation cluster in participant #3 (electrodes with black circles denote electrodes without any narrowband peaks in the power spectrum). (**d**) Beta frequency oscillation cluster in the same participant. (**e**) Mean power spectrum (blue) in log–log coordinates from an example electrode in participant #3. Black line denotes fitted 1/f line to the mean power spectrum in log–log coordinates using robust linear regression (see Materials and methods). (**f**) Normalized power spectrum (red) after removal of the 1/f background signal. This approach emphasizes narrowband oscillations as positive deflections. Stars (cyan) denote the narrowband peaks in the normalized power spectrum, identified as any local maximum greater than one standard deviation above the mean (see Materials and methods).

The online version of this article includes the following figure supplement(s) for figure 4:

**Figure supplement 1.** Oscillation clusters in participant #1.

**Figure supplement 2.** Oscillation clusters in participant #2.

*Figure 4 continued on next page*

Our next goal was to test whether the signals on these oscillation clusters behaved as traveling waves (Materials and methods). We first informally inspected the signals from the oscillation clusters and found that the instantaneous phases of their oscillations varied systematically with electrode locations, indicating the presence of traveling waves. *Figure 5a–f* shows an example of the systematic variation of the instantaneous phases across a subset of electrodes for both low-frequency theta-band and higher-frequency beta-band oscillation clusters in participant #3. Importantly, this example shows that a single group of electrodes could exhibit traveling waves at multiple frequencies simultaneously.

We used circular statistics (*Fisher, 1993*) to identify traveling waves at each time-point and electrode in a given oscillation cluster by measuring the phase of the oscillations on each electrode and testing for progressive shifts between neighboring electrodes. To do this systematically, for each electrode in a given oscillation cluster, we first identified a subcluster of neighboring electrodes, as those within a distance of 20 mm. Then, for each subcluster, we used a 2D circular–linear regression to characterize the spatial relation between the instantaneous phases of the oscillations and the positions of the electrodes (*Figure 5g*). Thus, this procedure models each subcluster's instantaneous phase distribution as a plane wave, estimated by finding the spatial gradient that provides the best fit to the propagation of phases across the nearby electrodes. The fitted phase gradient provides a quantitative estimate of the speed and direction of traveling wave propagation at each time-point (*Figure 5h*). With this procedure, we found that the instantaneous features of traveling waves varied across time (*Figures 5h–j and 6*). We computed the instantaneous wave-strength of the traveling waves as the mean proportion of phase variation explained by the circular–linear models across the cluster (Materials and methods).

To assess the statistical reliability of each cluster's traveling waves, we used statistics based on permutation testing. In this procedure, we compared each cluster's average wave-strength to the distribution of wave-strength values expected by chance under the null hypothesis that there was no relation between oscillation phase and electrode location within each cluster (Materials and methods). Because this shuffling procedure specifically evaluated the spatial arrangement of the recording electrodes, our results precluded the possibility that the results are driven by a specific subset of electrodes or their spatial arrangement. With this approach, we found that traveling waves were reliably present in this resting-state dataset, in all 9 of the participants. Significant traveling waves were present ~35% of the time on average across all clusters. Moreover, traveling waves were almost evenly present in both the left and right insula, in ~35% of timepoints in the left insula and ~35% in the right insula. Individual clusters varied in the prevalence of traveling waves, ranging from ~10% to ~87%. 93% of all insula oscillation clusters (26 out of 28 clusters) showed traveling waves at statistically significant levels. Overall, traveling waves had a median propagation speed of ~0.7 m/s, consistent with earlier work on traveling waves across the cortical surface (*Halgren et al., 2019*; *Stolk et al., 2019*; *Zhang et al., 2018*).

## Amplitude of oscillations is positively correlated with the strength of traveling waves

We next sought to investigate the relationship between the amplitude of the oscillations measured at individual electrodes and the strength of cluster-wide traveling waves, based on emerging findings from motor cortical local field potentials in non-human primates (*Denker et al., 2018*). Visual inspection revealed that such a link might be present, as we saw that the amplitude of oscillations from a cluster correlated with the simultaneous strength of traveling waves (*Figure 7a–f*).

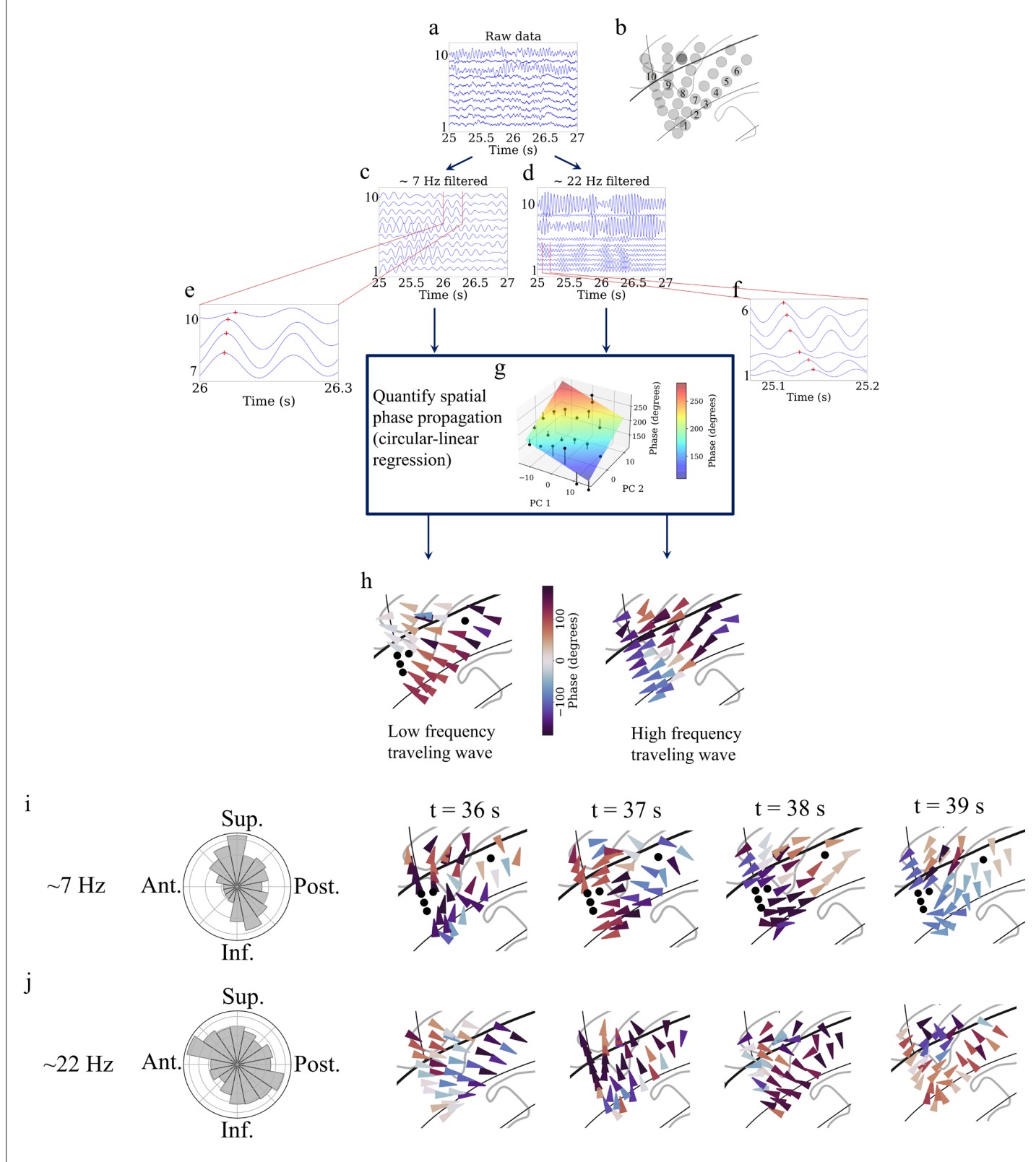

**Figure 5.** Detection of traveling waves in the human insula. We estimated Hilbert-transformed instantaneous phases of electrodes in the oscillation clusters, and if they vary systematically with the location of the electrodes then they constitute a traveling wave. We used a localized circular-linear regression approach to detect complex patterns of traveling waves in addition to linear traveling waves, in an oscillation cluster. This localized circular-linear regression approach estimated the direction of traveling waves for each electrode (Materials and methods). (**a**) Raw traces from electrodes as

*Figure 5 continued*

numbered in (**b**) for participant #3. (**c**) Filtered signals in the theta frequency band. (**d**) Filtered signals in the beta frequency band. (**e**) and (**f**) are zoomed-in versions of filtered signals in (**c**) and (**d**), respectively, to show (denoted as red plus signs) a spatially progressive variation of the phases across the electrodes (electrodes 7–10 for theta frequency and electrodes 1–6 for beta frequency). (**g**) Visual demonstration of circular-linear regression approach to detect traveling waves. 3D Talairach coordinates of electrodes were mapped to 2D coordinates using principal component analysis (PCA) (PC 1 and PC 2 denote the first and second principal components, respectively) and phases of the electrodes were fitted with these coordinates in the regression model. Solid circles denote the actual phases of the electrodes with the bars denoting the residuals between the actual phases and the predicted phases (Materials and methods). (**h**) Left panel: arrow plot showing instantaneous propagation direction of the theta traveling wave corresponding to *t*=25.225 s, right panel: arrow plot showing instantaneous propagation direction of the beta traveling wave corresponding to the same time-instant as in the left panel. (**i**) Panel 1 shows histogram of the instantaneous propagation directions across time for theta traveling wave. Panels 2–5: arrow plots showing variation of instantaneous propagation directions of the theta traveling waves corresponding to *t*=36 s, 37 s, 38 s, and 39 s, respectively. (**j**) Panel 1 shows histogram of the instantaneous propagation directions across time for the beta traveling wave. Panels 2–5: arrow plots showing variation of instantaneous propagation directions of beta traveling waves corresponding to *t*=36 s, 37 s, 38 s, and 39 s, respectively. Ant: anterior; Post: posterior; Sup: superior; Inf: inferior.

To systematically examine how local oscillation features correlated with traveling waves, at each timepoint we measured the instantaneous amplitude of the oscillations on each individual electrode in a cluster with the Hilbert transform and then measured the average amplitude across electrodes. Then, for each cluster, we computed the correlation over time between mean oscillation amplitude and traveling wave-strength using a permutation procedure to test statistical significance (Materials and methods). This analysis revealed that for ~71% of clusters (20 out of 28 clusters), there was a statistically significant correlation between amplitude and wave-strength (*Figure 7g–i*, *Supplementary file 1a*). This indicates that the strength of the traveling waves correlates with the amplitude of neuronal oscillations in the human insula.

Visual inspection also revealed temporal structure in this pattern, in which the amplitude of oscillations in a cluster peaked before wave-strength (*Figure 7b–c*). To quantify the temporal delay between the amplitude and wave-strength time-series, we carried out a cross-correlation analysis (*Rabiner and Gold, 1975*). This analysis revealed that oscillation amplitude increases prior to increase in wave-strength, with a mean shift of 8 ms for theta traveling waves and 4 ms for beta traveling waves. Together, these results suggest a putative causal role of higher amplitude oscillations in driving stronger traveling waves.

## Theta and beta traveling waves are temporally uncorrelated

Finally, we investigated the relationship between the lower-frequency theta traveling waves and higher-frequency beta traveling waves, based on emerging findings that showed that low-frequency alpha and higher-frequency beta traveling waves are uncorrelated or weakly correlated with each other in the sensorimotor cortex (*Stolk et al., 2019*). To explore the relationship between the theta and beta traveling waves in our dataset, we carried out correlation analyses examining the interrelations between the instantaneous strength, phase, and oscillatory power of traveling waves between low and high frequencies for any oscillation clusters that showed significant traveling wave clusters at multiple frequencies. To test the statistical significance of links between the features of low-frequency and high-frequency traveling waves, we circularly shuffled the values of the higher-frequency and lower-frequency traveling waves relative to each other to build a surrogate distribution of correlation values corresponding to the null hypothesis that the features of simultaneous traveling waves at different frequencies were uncorrelated over time.

Overall, this analysis showed that, overwhelmingly, when individual electrodes participated in multiple traveling waves at different frequencies, these effects were largely independent. Across all clusters that showed beta and theta traveling waves, the instantaneous strengths of these traveling waves were uncorrelated (Pearson *r*, p>0.05) for all but two clusters (*Supplementary file 1b*). In addition to testing for correlations in wave-strength, to test for more fine-grained links between different traveling waves, we compared how the instantaneous phase and power of slower theta-band traveling waves related to the strength of the beta traveling waves. However, we found no such links (Pearson *r*, p>0.05) (*Supplementary file 1b*). Together, these results suggest that across the insula, low-frequency theta and higher-frequency beta traveling waves operate independently.

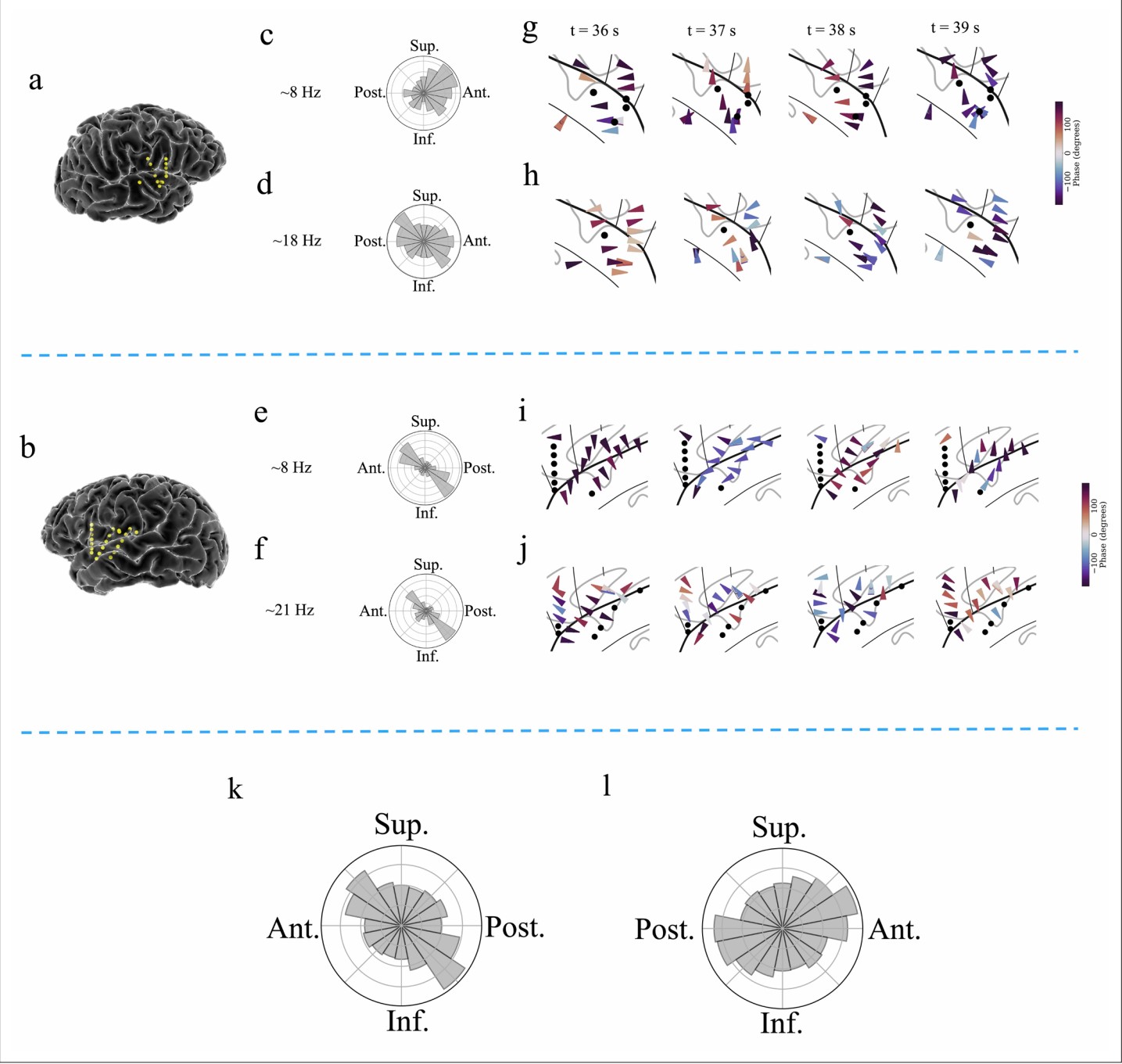

**Figure 6.** Examples of traveling waves in the human insula. Traveling waves changed their directions with respect to time. Insula electrodes visualized in a brain surface plot in (**a**) participant #6 and (**b**) participant #7. (**c**) and (**d**) are histograms of the instantaneous propagation directions across time for theta (~8 Hz) and beta (~18 Hz) traveling waves in participant #6. (**e**) and (**f**) are the same as in (**c**) and (**d**), respectively, for participant #7. (**g**) Panels 1–4: arrow plots showing variation of instantaneous propagation direction of theta traveling waves corresponding to *t*=36 s, 37 s, 38 s, and 39 s, respectively, in participant #6, visualized in a zoomed-in version of flat brain. (**h**) Panels 1–4: arrow plots showing instantaneous propagation directions of beta traveling waves corresponding to *t*=36 s, 37 s, 38 s, and 39 s, respectively, in participant #6, visualized in a zoomed-in version of flat brain. (**i**) and (**j**) are same as in (**g**) and (**h**), respectively, for participant #7. (**k**) and (**l**) are the histograms of the instantaneous propagation directions of traveling waves across time, clusters, and participants for left and right insula, respectively. Solid circles represent electrodes without narrowband oscillations. Ant: anterior; Post: posterior; Sup: superior, Inf: inferior.

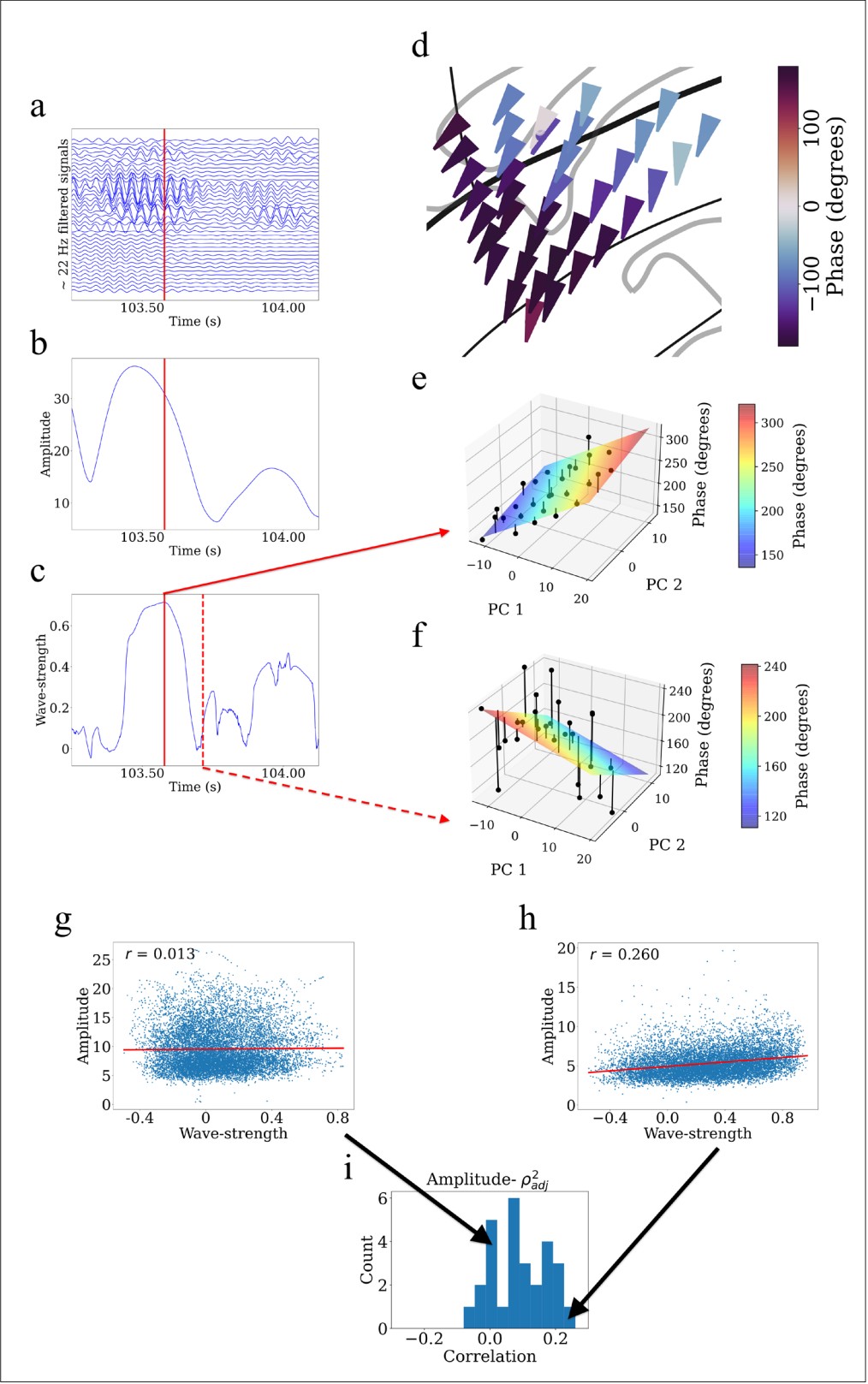

**Figure 7.** Example showing that higher wave-strength values of traveling waves correspond to higher Hilbert-transformed amplitude of signals in participant #3. (**a**) Filtered signals in the beta frequency band from all electrodes in participant #3 in time interval 103.25 s to 104.125 s. (**b**) Average amplitude across all electrodes in the same time interval. We defined the average amplitude for each oscillation cluster to be the average of the Hilbert-

*Figure 7 continued on next page*

*Figure 7 continued*

transformed instantaneous amplitudes of filtered signals of all electrodes. (**c**) Wave-strength, averaged across all electrodes, in the same time interval. Higher wave-strength corresponds to higher amplitudes of electrodes in (**b**) (But observe that the amplitude peaks earlier than the wave-strength; see Results). Time-instant corresponding to the local maximum of wave-strength in (**c**) is denoted by the red lines in (**a**), (**b**), and (**c**). (**d**) Arrow plot showing instantaneous propagation direction of electrodes for the beta traveling wave at the time-instant corresponding to the red line in (**c**). Note the smooth variation of phases across space, which results in the high value of wave-strength in (**c**). (**e**) Circular-linear regression fit at the time-instant corresponding to the red line in (**c**). Solid circles denote the actual phases of the electrodes with the bars denoting the residuals between the actual phases and the predicted phases. Note the relatively low values of the residuals indicated by the low heights of the bars. (**f**) Circular-linear regression fit at a time-instant (denoted as dotted red line in (**c**)) for which the wave-strength is relatively low. Note the relatively high values of the residuals. (**g–i**) Strength (denoted as $\rho^2_{adj}$) of traveling waves is positively correlated with Hilbert-transformed amplitude of signals. We estimated the average (averaged across electrodes) Hilbert-transformed instantaneous amplitudes of signals of the electrodes for each oscillation cluster and estimated their correlation with the wave-strength of traveling waves. We used a circular-shuffling procedure to test the significance of correlations where we circularly shuffle the wave-strength values with respect to the average amplitude and estimate the correlation on this shuffled data to build a distribution of surrogate correlation values against which the observed correlation was tested (p<0.05) (Materials and methods). (**g**) Amplitude- $\rho^2_{adj}$ scatter plot for cluster 1 of participant #7 where there was no statistically significant correlation (p>0.05). (**h**) Amplitude- $\rho^2_{adj}$ scatter plot for cluster 2 of participant #8 where there was a statistically significant correlation (p<0.05). (**i**) Histogram of correlations across all clusters (also see *Supplementary file 1a*).

## Discussion

Here, we used a unique resting-state iEEG dataset of 10 participants who underwent treatment for intractable epilepsy, and electrodes were sampled along the extent of the insula with implants in both the anterior and posterior subdivisions of the insula across participants. With this dataset, we carried out a comprehensive analysis of frequency and power gradients along the oscillations we measured along the AP and SI axes in the insula, for the first time. Using spectral analyses, we found that oscillations in the theta and beta bands are the most prominent spontaneous oscillations in the insula (*Figure 1*), and these oscillations are largely organized along the AP axis. Notably, there were no 9–15 Hz oscillations in the insula (*Figure 1*). The left and right insula showed convergent patterns of anterior-to-posterior spectral power gradients in the beta band (~15–32 Hz). Intriguingly, theta (~6–9 Hz) oscillations were heterogeneous across hemispheres and showed a gradient of frequency that matched the insula's hierarchical organization. In the left insula, theta oscillations showed a posterior-to-anterior frequency gradient and an anterior-to-posterior power gradient. However, in the right insula, theta oscillations showed a superior-to-inferior power gradient, but no frequency gradient.

We next identified traveling waves corresponding to the theta and beta oscillations. We identified spatially contiguous clusters of electrodes at theta and beta frequency bands and then used a novel, localized circular–linear regression approach to detect traveling waves, assuming that the relative phases of the oscillation clusters exhibit a linear relationship with electrode locations locally. In addition to detecting traveling waves in most of these participants, we also found that the amplitudes of neuronal oscillations are positively correlated with the strength of traveling waves in the insula. Furthermore, this analysis also revealed that theta and beta oscillatory traveling waves in the insula are uncorrelated with each other. These findings provide a more complete understanding of spontaneous neuronal oscillations in the human insula by showing that this structure shows multiple independent traveling waves during rest.

The insula is a key brain region that was shown with fMRI to play a crucial role in a wide range of adaptive human behaviors (*Menon and Uddin, 2010*; *Singer et al., 2009*). However, its electrophysiological foundations have remained unexplored due to the lack of sufficient electrode coverage. Recently, with the increased popularity of stereo EEG recording technology (*Abou-Al-Shaar et al., 2018*), we have much more access to deep brain structures such as the insula in epilepsy patients and can explore the electrophysiological foundations, subsecond dynamics, and spatiotemporal gradients in this area for the first time. Crucially, our findings of spontaneous gradients and traveling waves in the insula suggest a fundamental new principle that could be used by the insula to link its processing to widespread cortical areas.

# Hierarchical organization of spectral frequency and power of theta and beta oscillations in the human insula

The first goal of our study was to detect frequency and power gradients in the human insula. Our analysis of the hierarchical organization of theta oscillations in the insula revealed that, in the left insula, theta oscillations have a posterior-to-anterior frequency gradient and an anterior-to-posterior power gradient. This finding is consistent with the global frequency and power gradients found in the human neocortex with MEG (*Mahjoory et al., 2020*) and iEEG (*Zhang et al., 2018*), as well as the hippocampal frequency gradients seen in iEEG recordings in humans during virtual spatial navigation (*Goyal et al., 2020*). Our findings show that the frequency and power gradients observed in other cortical areas also extend to the insula. Our findings also corroborate previous resting-state fMRI (*Deen et al., 2011*; *Faillenot et al., 2017*) and diffusion MRI (*Menon et al., 2020*) findings in humans, which identified distinct functional subdivisions in the insula based on differential patterns of intrinsic functional connectivity and extended them to electrophysiological data. These findings also converge with work showing the presence of spindle-shaped von Economo neurons in AI (*Cobos and Seeley, 2015*; *Evrard et al., 2012*; *Seeley et al., 2012*; *von Economo, 1926*; *Watson et al., 2006*), which form the agranular part of the insula. The AI lacks external (layer II) and internal granular (IV) layers, which may subserve its special role of integrating information across wide-spread neuronal networks (*Mesulam and Mufson, 1985*; *Morel et al., 2013*; *Wylie et al., 2015*). By showing distinctive patterns of oscillations for the AI versus PI that match broader cortex-wide patterns, it supports the notion of the insula as a communication hub and suggests that distinctive patterns of oscillations play a role in allowing it to engage with other regions, such as the DMN and FPN, across a range of cognitive and affective tasks (*Cai et al., 2016*; *Cai et al., 2021*; *Chen et al., 2016*; *Lamm and Singer, 2010*; *Sridharan et al., 2008*).

In our previous study involving the AI using resting-state iEEG recordings in humans, we showed that low-frequency delta (~0.5–4 Hz) oscillations underlie higher directed causal information flow from the AI to other brain regions such as the DMN and the FPN (*Das and Menon, 2020*). Taken together with our current results, lower-frequency and higher power oscillations in the AI compared to PI seem to be a putative mechanism that enables the AI to maintain stability and flexibility with other large-scale brain networks (*Cai et al., 2016*; *Cai et al., 2021*; *Chen et al., 2016*; *Lamm and Singer, 2010*; *Sridharan et al., 2008*).

However, surprisingly, in the right insula, we found no AP theta gradients in frequency and only weak AP theta gradients in power. Moreover, right insula theta oscillations showed a superior-to-inferior power gradient, but no frequency gradient. This finding is consistent with structural MRI studies in humans that showed a left-insula-dominated connectivity profile with the prefrontal and frontal brain regions (*Jakab et al., 2012*). This left insula dominance has also been confirmed by other structural MRI studies (*Chiarello et al., 2013*; *Van Essen et al., 2012*). Thus, our findings of left-insula-dominated theta frequency and power gradients converge with the structural MRI studies and provide novel evidence for the electrophysiological basis of this structural asymmetry. These structural MRI studies have further shown that the anterior, rather than the posterior, insula in the left hemisphere is more involved in semantic decisions (*Chiarello et al., 2013*), consistent with the frequency and power gradients that we found along the AP axis of the insula. We hypothesize that these spontaneous frequency and power gradients in the left insula also play a key role in modulating human behavior. With the structural MRI findings, these results suggest that oscillatory gradients may help the AI in integrating information from other parts of the insula and projecting them to other parts of the cortex for further processing, given its widespread cortical connectivity. Future studies with denser sampling of electrodes in both the AI and the PI are needed to further probe the hemispheric heterogeneity of low-frequency gradients that we observed.

We next investigated the hierarchical organization of beta oscillations. This analysis revealed that both the left and right insula have anterior-to-posterior spectral power gradients. Gradients in the human brain in previous iEEG studies have predominantly been found in low-frequency oscillations (*Das and Menon, 2020*; *Goyal et al., 2020*; *Zhang et al., 2018*). However, consistent with our results, some recent electrophysiological studies have also found a crucial role for beta oscillations in feedback signaling in both non-human primate and human brains (*Bastos et al., 2015*; *Das and Menon, 2021*). Beta oscillations have been hypothesized to underlie directed causal information flow from one brain region to another and may contribute to transitioning latent neuronal ensembles

into 'active' representations (*Spitzer and Haegens, 2017*), as well as the subsequent maintenance of information in cell assemblies (*Engel and Fries, 2010*). Thus, higher beta power in the AI may facilitate communication with other large-scale brain networks via feedback signaling. In summary, the frequency gradients were most prominent in the left insula, whereas the power gradients were present in both hemispheres. We hypothesize that the left hemisphere insula communicates via both frequency and power gradients, whereas the right hemisphere insula communicates via power gradients. Multiplexing of frequency and power gradients and hemispheric asymmetry may help the AI maintain a powerful and flexible role in the cognitive control mechanisms as hypothesized previously in human fMRI studies (*Cai et al., 2016*; *Cai et al., 2021*; *Chen et al., 2016*; *Lamm and Singer, 2010*; *Sridharan et al., 2008*).

## Oscillation clusters in the human insula are spatially localized traveling waves

Our second overarching goal was to detect traveling waves of theta and beta frequency oscillations. This was inspired by previous work on computational modeling of weakly coupled oscillators that has shown that traveling waves can naturally emerge from linear frequency gradients (*Ermentrout and Kopell, 1984*). Therefore, we examined whether the frequency and power gradients that we observed also exhibited traveling waves. We found that traveling waves are present ~35% of the time in the human insula across all clusters. Our findings converge with resting-state rodent (*Matsui et al., 2016*) and human (*Bahramisharif et al., 2013*; *Halgren et al., 2019*) electrophysiology studies as well as more recent resting-state human fMRI studies (*Raut et al., 2021*), which detected traveling waves of spontaneous neuronal oscillations and extend these findings to the human insula. These findings also converge with recent findings that have detected relatively sparse traveling waves in the rodent somatosensory cortex (*Moldakarimov et al., 2018*), visual cortex of non-human primates (*Davis et al., 2020*), and iEEG recordings in humans (*Zhang et al., 2018*). Intriguingly, previous resting-state iEEG studies in humans (*Bahramisharif et al., 2013*; *Halgren et al., 2019*) found traveling waves in the alpha band only, which we specifically did not observe here. However, the role of traveling waves of spontaneous beta oscillations in the human cortex is relatively unknown, although traveling waves have been detected during sleep spindles, which tend to occur in the beta frequency range (~11–15 Hz) (*Muller et al., 2016*). Our finding of traveling waves in the beta frequency band in the human insula suggests a role for beta band for communicating information within the insula and also dynamically coordinating this information with other brain networks. This result may explain the role of the insula in a wide range of adaptive human behaviors (*Menon and Uddin, 2010*), which has been hypothesized as an important role of beta frequency oscillations (*Betti et al., 2021*).

Lack of sufficient electrode placements (*Figure 4—figure supplements 1–9*) simultaneously along both the AP and dorsal–ventral portions of the insula in individual participants precluded a complete population-level analysis of the propagation directions of the traveling waves. Nevertheless, analysis of the propagation directions of the traveling waves in selected participants with adequate electrode sampling (*Figures 5–6*) revealed directional modulation of insular traveling waves; however, they were different across participants. Future studies with adequate sampling of electrodes in the insula of individual participants are needed to rigorously analyze the instantaneous propagation directions of traveling waves and to relate them directly to the frequency and power gradients that we observed in the insula.

## Features of theta and beta traveling waves in the human insula

Finally, we also investigated the relationship between the amplitude of low-frequency and high-frequency oscillations and the strength of simultaneous traveling waves. For ~71% of clusters, we found a statistically significant correlation between oscillation amplitude and wave-strength, and this correlation was positive in almost every case (19 of these 20 clusters). Cross-correlation analysis revealed that the median temporal delay between amplitude and wave-strength is ~8 ms for theta traveling waves and ~4 ms for beta traveling waves, with the wave-strength time-series leading the amplitude time-series in both cases. Together, these time shifts suggest a putative causal role of higher amplitude oscillations underlying higher wave-strength traveling waves in the human insula. Finally, we investigated the relationship between traveling waves at theta and beta frequencies. Recent findings in human electrocorticography studies have shown that low-frequency alpha and high-frequency beta

traveling waves are temporally uncorrelated or weakly correlated with each other in the sensorimotor cortex (*Stolk et al., 2019*). Our analysis confirmed that this pattern also persists in the human insula.

Interestingly, our findings also converge on a recent study (*Marks et al., 2021*) which used a large dataset of iEEG recordings from 164 participants showing temporal independence of low-frequency and high-frequency oscillations during the memory encoding period of a verbal episodic memory task. This pattern suggests that putatively orthogonal frequency channels may support spatiotemporal information transfer within the insula. We hypothesize that these spatiotemporally orthogonal channels enable the insula to operate at the apex of cognitive control networks observed in the human brain (*Cai et al., 2021*) and initiate dynamic switching with the DMN and FPN, which has been reported across a wide range of cognitive and affective tasks (*Cai et al., 2016*; *Cai et al., 2021*; *Chen et al., 2016*; *Lamm and Singer, 2010*; *Sridharan et al., 2008*).

## Conclusions

The insula plays a critical role in a range of adaptive human behaviors (*Menon and Uddin, 2010*; *Singer et al., 2009*) and insula dysfunction is prominent in many psychiatric and neurological disorders (*Jilka et al., 2014*; *King-Casas et al., 2008*; *Sha et al., 2019*). Crucially, the intrinsic connectivity of the insula displays close correspondence with task-related coactivation patterns, and this correspondence has allowed the intrinsic and task-related connectivity associated with the insula to be demarcated and studied under a common framework (*Sridharan et al., 2008*; *Supekar and Menon, 2012*). Using a unique dataset of resting-state human iEEG recordings from 10 participants and 239 electrodes in the insula, we probed its hierarchical organization and spatiotemporal dynamics and, for the first time, detected traveling waves of neuronal oscillations in theta and beta frequency bands in the human insula. Our findings significantly advance the understanding of the neurophysiological basis of spontaneous neuronal oscillations in the insula and, more broadly, of the foundational mechanisms underlying large-scale cortical hierarchies in the brain. Crucially, our findings of spontaneous gradients and traveling waves in the insula suggest a fundamental new principle that could be used by the insula to link its processing to widespread cortical areas.

# Materials and methods

## Participants

Intracranial electroencephalography (iEEG) recordings were acquired from 10 patients (5 females; 9 right-handed, 1 ambidextrous) with drug-intractable epilepsy. The average age of the patients was ~31 with an age of seizure onset between 9 months old and 46 years old. All decisions regarding the location and coverage of the iEEG probes were based solely on clinical criteria. The Baylor College of Medicine Institutional Review Board approved the placement of all electrodes (IRB-18112). All patients provided informed consent before participating. The insula was not the origin of seizure onset for any patients in this study.

## Electrophysiological recordings and preprocessing

We used resting state electrophysiological recordings from participants performing a passive fixation resting state task, where they focused on a white crosshair at the center of a black background on a computer monitor for ~5 min duration. Data was recorded from 239 iEEG electrodes using a neural signal processor (Blackrock Microsystems). The spacing between neighboring electrodes varied from 3.5 mm to 5.53 mm (center-to-center). The data was amplified, band-pass filtered (0.3 Hz–7.5 kHz), and digitized at 2 kHz. For analysis of spectral frequency and power gradients, we used bipolar referencing (iEEG electrodes re-referenced to their closest neighbors) and for analysis of traveling waves, we used common average referencing (iEEG electrodes re-referenced to the average signal of all electrodes). Automatic data rejection was performed with EEGLAB based on 3× the SD of spectra from 1 to 50 Hz in order to remove data with artifactual voltage deflections and epileptiform discharges (*Delorme and Makeig, 2004*). Each subject contributed an average of 23.9 (±2.43) electrodes, with 12.5 (±4.04) in the left insula and 11.4 (±2.55) in the right insula. We also report, in *Figure 1—figure supplement 2* the anatomical distribution of electrodes across the insula. This shows wide coverage of electrodes across both the anterior (which includes the short-anterior, short-middle, and short-posterior gyri subdivisions) and posterior (which includes the long-anterior and long-posterior subdivisions) regions

as well as the apex region of the insula. Four of the ten subjects had bilateral insula coverage. A zero-phase notch filter was used to remove 60 Hz line noise and harmonics.

## Electrode localization

An automatic cortical reconstruction was performed on the preoperative T1-weighted MRI Free-Surfer (https://surfer.nmr.mgh.harvard.edu/). All iEEG electrodes were localized by co-registering each patient's presurgical structural T1-weighted MRI with their postsurgical CT images (*Fischl, 2012*) using the Functional Magnetic Resonance Imaging for the Brain Software Library's (FMRIB's) Linear Image Registration Tool (FLIRT) (*Jenkinson et al., 2002*; *Jenkinson and Smith, 2001*). All neuroimages were coregistered to standard MNI 152 space in an iterative semiautomated manner. With ITK-SNAP (version 3.8.0) (*Yushkevich et al., 2006*), the presurgical T1-weighted MRI was manually aligned to the standard MNI 152 atlas image before a rigid registration was performed. A global affine transformation was subsequently applied to refine the coregistration. Bilateral insula regions from the MNI 152 atlas were dilated by five voxels and combined to create a mask, which was then used in the final registration step. A local affine transformation with this mask was applied three times to optimize insular coregistration from patient space to MNI 152 space. Mutual information was used as the similarity metric for all registrations. The accuracy of each registration was manually verified before the resulting transformation matrix was applied to the postsurgical CT images. With the postsurgical CT registered to MNI 152 space, the resulting electrode coordinates were localized and extracted using BioImage Suite (*Joshi et al., 2011*). An expert rater (B. S.) examined the fused MRI to CT images and determined whether each contact was in white or gray matter based on its location on the brain slice. Contacts labeled 'white matter' had no voxels within 3 mm of gray matter. The vast majority (~97%) of all contacts in this study were in insula gray matter. Additionally, to verify the results of our co-registration approach, electrodes located in the insula were first visually identified by a neurosurgeon and further verified using the latest Desikan–Killiany (DK) atlas, available in the Freesurfer software (*Desikan et al., 2006*; *Fischl, 2012*). Anatomical parcellation of the insula for image generation was performed using the DK atlas.

## Analysis of frequency and power gradients

Power spectral density (PSD) was computed for each electrode via wavelet convolution. Each signal, $x(t)$, from a given electrode, was convolved with a Morlet wavelet defined by

$$\omega(t, f) = A * e^{\frac{-t^2}{2\sigma_t^2}} * e^{i2\pi f_t},$$

where $t$ is time, $f$ is frequency, and $\sigma_t$ is wavelet duration. $A$ is normalized amplitude, such that $A = \sigma_t \sqrt{\pi}^{-1/2}$. The number of cycles in each wavelet function was 6. The resting-state recordings were 5 min duration and analyzed from 1 to 50 Hz in 0.1 Hz intervals (491 frequencies). Frequency bands for spatial gradient analyses were selected from the average of all electrodes using an oscillation detecting procedure called MODAL (*Watrous et al., 2018*). MODAL detects frequency bands where power exceeds one SD above the 1/f background spectrum. Across the insula, theta (~6–9 Hz) and beta (~15–32 Hz) oscillations were detected, however, we did not observe significant oscillation clusters in the ~9–15 Hz frequency band (*Figure 1*). Peak frequencies and the corresponding power within theta and beta bands were selected by applying a robust (bisquare) linear fit to the power spectral density function and then locating the frequencies that corresponded to the maximum amplitude above the residuals.

## Traveling waves analysis

We adopted a novel method to identify traveling waves in the insula corresponding to neuronal oscillations in each participant individually. These consisted of two primary steps: (i) identification of spatially contiguous clusters of electrodes at theta and beta bands and (ii) identification of systematic spatial variation of instantaneous oscillatory phase across the electrodes for each cluster, which we defined as a traveling phase wave. These steps are detailed below.

## Identification of spatial clusters

To characterize propagating traveling waves, we first sought to identify spatial clusters of electrodes with narrowband oscillations at the theta and beta bands. We adopted methods similar to our previous

approach (*Zhang et al., 2018*). This algorithm accounted for several complexities of human brain oscillations measured with iEEG signals, including differences in electrode positions across subjects and variations in oscillation frequencies across individuals.

Similar to the PSD analysis above, we first used Morlet wavelets to compute the power of the neuronal oscillations throughout the resting-state recordings of ~5 min duration at 200 frequencies logarithmically spaced from 3 to 40 Hz. To identify narrowband oscillations at each site, we fit a line to each patient's mean power spectrum in log–log coordinates using a robust linear regression as above (*Figure 4*). We then subtracted the actual power spectrum from the regression line. This normalized power spectrum removes the 1/f background signal and emphasizes narrowband oscillations as positive deflections. We identified narrowband peaks in the normalized power spectrum as any local maximum greater than one standard deviation above the mean.

Next, we implemented a spatial clustering algorithm to identify oscillation clusters, which we defined as contiguous groups of electrodes in each subject that exhibited narrowband oscillations at theta (~4–8 Hz) and beta (~12–30 Hz) frequencies. To test whether the electrodes comprised a spatially contiguous group, we created a pairwise-adjacency matrix to judge their spatial proximity. This matrix indicated whether each electrode pair was separated by less than 20 Talairach units (20 mm). Finally, we used this adjacency matrix to identify mutually connected spatial clusters of electrodes by computing the connected components of this graph (*Tarjan, 1972*). We included in our analyses only clusters with at least four connected electrodes. The clustering analysis was performed based only on the proximity of the electrodes with respect to frequency and distance and not by their position along the anteroposterior axis. Consequently, we do not distinguish electrodes or clusters based on their position within AI or PI for the traveling waves analysis.

## Identification of traveling waves

We next identified traveling waves corresponding to the spatially contiguous groups of electrodes oscillating at theta and beta frequencies (*Figure 4*, *Figure 4—figure supplements 1–9*) that showed oscillations moving progressively across the cortex. Quantitatively, a traveling phase wave can be defined as a set of simultaneously recorded neuronal oscillations at the same frequency band whose instantaneous phases vary systematically with the location of the recording electrode. We used a localized circular–linear regression approach, assuming that the relative phases of the oscillation clusters exhibit a linear relationship with electrode locations *locally*. This local circular-linear fitting of phase-location can detect complex patterns (*Ermentrout and Kleinfeld, 2001*; *Muller et al., 2016*) of traveling waves in an oscillation cluster in addition to linear traveling waves. This is different than our previous approach (*Zhang et al., 2018*) in which we carried out circular-linear fitting of phase-location with one direction for an entire oscillation cluster at a time, which assumed that phase gradients exhibit a fixed linear relationship with electrode location across the entire cluster.

To identify traveling waves from the phases of each oscillation cluster, we first measured the instantaneous phases of the signals from each electrode of a given cluster by applying a third-order Butterworth filter at the cluster's narrowband peak frequency (bandwidth [$f_p \times 0.85$, $f_p/0.85$] where $f_p$ is the peak frequency). We used a Hilbert transform on each electrode's filtered signal to extract the instantaneous phases.

We used circular statistics to identify traveling waves of phase progression for each oscillation cluster at each time point (*Fisher, 1993*). For each spatial phase distribution, we used 2D localized circular–linear regression to assess whether the observed phase pattern varied linearly with the electrode's coordinates in 2D. In this regression, for each electrode in a given oscillation cluster, we first identified the neighboring electrodes located within 20 mm distance of the given electrode, constituting a sub-cluster of the given cluster. We included in our subsequent analyses only sub-clusters with at least four electrodes. Let $x_i$ and $y_i$ represent the 2D coordinates and $\theta_i$ the instantaneous phase of the $i$th electrode. The 2D coordinates $x_i$ and $y_i$ are determined by projecting the 3D Talairach coordinates for each sub-cluster into the best-fitting 2D plane using the principal component analysis. We projected the electrode coordinates into a 2D space to simplify visualizing and interpreting the data.

We used a 2D circular-linear model:

$$\hat{\theta}_i = (ax_i + by_i + \theta) \bmod 360°,$$

where $\hat{\theta}_i$ is the predicted phase, a and b are the phase slopes corresponding to the rate of phase change (or spatial frequencies) in each dimension, and θ is the phase offset. We converted this model to polar coordinates to simplify fitting. We define α=atan2(b, a) which denotes the angle of wave propagation and $\xi = \sqrt{a^2 + b^2}$ which denotes the spatial frequency. Circular–linear models do not have an analytical solution and must be fitted iteratively (**Fisher, 1993**). We fitted α and $\xi$ to the distribution of oscillation phases at each time point by conducting a grid search over $\alpha \in \left[0^0,\ 360^0\right]$ and $\xi$ in increments of $5^0$ and $0.5^0$/mm, respectively (Note that $\xi$ =32.5 corresponds to the spatial Nyquist frequency of $32.5^0$ /mm corresponding to the highest spacing between neighboring electrodes of 5.53 mm.). The model parameters (a = ξcos(α) and b = ξsin(α)) for each time point are fitted to most closely match the phase observed at each electrode in the subcluster. We computed the goodness of fit as the mean vector length $\bar{r}$ of the residuals between the predicted ($\hat{\theta}_i$) and actual ($\theta_i$) phases (**Fisher, 1993**),

$$\bar{r} = \sqrt{\left[\frac{1}{n}\sum_{i=1}^{n} cos\left(\theta_i - \hat{\theta}_i\right)\right]^2 + \left[\frac{1}{n}\sum_{i=1}^{n} sin\left(\theta_i - \hat{\theta}_i\right)\right]^2},$$

where *n* is the number of electrodes. The selected values of $\alpha$ and $\xi$ are chosen to maximize $\bar{r}$. This procedure is repeated for each sub-cluster of a given oscillation cluster. To measure the statistical reliability of each fitted traveling wave, we examined the phase variance that was explained by the best fitting model. To do this, we computed the circular correlation $\rho_{cc}$ between the predicted ($\hat{\theta}_i$) and actual ($\theta_i$) phases at each electrode:

$$\rho_{cc} = \frac{\sum_{i=1}^{n} sin\left(\theta_i - \bar{\theta}\right) sin\left(\hat{\theta}_i - \bar{\hat{\theta}}\right)}{\sqrt{\sum_{i=1}^{n} sin^2\left(\theta_i - \bar{\theta}\right) \sum_{i=1}^{n} sin^2\left(\hat{\theta}_i - \bar{\hat{\theta}}\right)}},$$

where bar denotes averaging across electrodes. Finally, we applied an adjustment for number of fitted model parameters:

$$\rho_{adj}^2 = 1 - \frac{\left(1 - \rho_{cc}^2\right)\left(n - 1\right)}{n - k - 1},$$

where *k* is the number of independent regressors (k=3 in this case). We refer to $\rho_{adj}^2$ as the wave-strength of the traveling wave (**Zhang et al., 2018**) as it quantifies the strength of the traveling wave (note that $\rho_{adj}^2$ has been referred to as phase gradient directionality (PGD) in some prior studies [**Muller et al., 2016**; **Rubino et al., 2006**; **Zhang et al., 2018**]).

## Statistical analysis

We used linear mixed-effects models to test the statistical significance of the frequency and power gradients. For analysis of frequency gradients, the peak frequency of each electrode was the dependent variable and the y-axis (AP) and z-axis (SI) coordinates of the electrodes were used as independent variables. Furthermore, a categorical independent variable, indicating the subject that contributed to the given frequency, was also included to account for variance across subjects. Interaction terms between all independent variables were included in the models as well. The spatial models were constructed as follows,

$$EPHYS = \beta_0 + \beta_{AP} + \beta_{SI} + \beta_{SUBJ} + \left(\beta_{AP} * \beta_{SI}\right) + \left(\beta_{AP} * \beta_{SUBJ}\right) + \left(\beta_{SI} * \beta_{SUBJ}\right),$$

where *EPHYS* can represent spectral frequency, or spectral power. Left and right hemisphere data were analyzed separately because they are known to play different functional roles that may contribute to differences in oscillatory organization (**Sridharan et al., 2008**). False discovery rate (FDR) correction was applied to all resulting p-values to control for multiple comparisons.

Surrogate analysis was used to test statistical significance of the detected traveling waves. The coordinates of the electrodes with respect to the Hilbert-transformed phases were shuffled so that the contiguous spatial variation of the phases is destroyed. This ensured that our statistical test specifically evaluated the spatial arrangement of the recording electrodes. For each iteration of this procedure, the circular-linear regression analysis was repeated on this shuffled data to build

a distribution of surrogate $\rho^2_{adj}$ values against which the observed $\rho^2_{adj}$ was tested (p<0.05). Similarly, to test the statistical significance of the correlations for amplitude-$\rho^2_{adj}$, $\rho^2_{adj}$(low)- $\rho^2_{adj}$(high), phase (low)-$\rho^2_{adj}$(high), and power (low)-$\rho^2_{adj}$(high), the values from one time-series were circularly shuffled with respect to the other so that the instantaneous correlation between the two time-series is destroyed. We used standard Pearson correlation coefficients to test the correlations for amplitude-$\rho^2_{adj}$, $\rho^2_{adj}$(low)- $\rho^2_{adj}$(high), and power (low)- $\rho^2_{adj}$(high). We used circular-linear correlation coefficients (*Johnson and Wehrly, 1977*; *Mardia, 1976*) to compare the correlations involving phase (i.e. phase (low)- $\rho^2_{adj}$(high)).

## Acknowledgements

We thank Drs. Uma Mohan and Honghui Zhang for help in traveling waves analysis. This research was supported by NIH grants R01-MH127006 and K01-MH116364 to K.B. and B.M., an NSF CAREER award to J.J., and grants from McNair Foundation and Dana Foundation to S.A.S.

## Additional information

### Funding

| Funder | Grant reference number | Author |
| --- | --- | --- |
| National Institutes of Health | R01-MH127006 | Brian A Metzger Kelly Bijanki |
| National Institutes of Health | K01-MH116364 | Brian A Metzger Kelly Bijanki |
| National Science Foundation | CAREER Award | Joshua Jacobs |
| McNair Foundation | McNair Foundation | Sameer A Sheth |
| Dana Foundation | Dana Foundation | Sameer A Sheth |

The funders had no role in study design, data collection and interpretation, or the decision to submit the work for publication.

### Author contributions

Anup Das, Conceptualization, Formal analysis, Investigation, Methodology, Writing – original draft, Writing – review and editing; John Myers, Conceptualization, Formal analysis, Investigation, Methodology, Writing – review and editing; Raissa Mathura, Ben Shofty, Chengyuan Wu, Methodology, Writing – review and editing; Brian A Metzger, Kelly Bijanki, Funding acquisition, Methodology, Writing – review and editing; Joshua Jacobs, Sameer A Sheth, Conceptualization, Funding acquisition, Investigation, Supervision, Writing – review and editing

### Author ORCIDs

Anup Das http://orcid.org/0000-0002-8897-7021
John Myers http://orcid.org/0000-0002-2412-7124
Kelly Bijanki http://orcid.org/0000-0003-1624-8767
Joshua Jacobs http://orcid.org/0000-0003-1807-6882

### Ethics

Human subjects: All decisions regarding the location and coverage of the iEEG probes were based solely on clinical criteria. The Baylor College of Medicine Institutional Review Board approved placement of all electrodes (IRB-18112). All patients provided informed consent before participating.

### Decision letter and Author response

Decision letter https://doi.org/10.7554/eLife.76702.sa1
Author response https://doi.org/10.7554/eLife.76702.sa2

# Additional files

## Supplementary files

• Supplementary file 1. Supplementary File for "Spontaneous Neuronal Oscillations in the Human Insula are Hierarchically Organized Traveling Waves".

(a). Wave-strength (denoted as $\rho^2_{adj}$) of traveling waves is positively correlated with Hilbert-transformed amplitude of signals. We estimated the correlation between the average amplitude of signals of the electrodes and the wave-strength of traveling waves for each oscillation cluster. We used a circular-shuffling procedure to test the significance of correlations where we circularly shuffle the wave-strength values with respect to the average amplitude and estimate the correlation on this shuffled data to build a distribution of surrogate correlation values against which the observed correlation was tested (p<0.05) (Materials and methods). We found that for ~71% of clusters (20 out of the 28 clusters), there was a statistically significant correlation between amplitude and wave-strength. All, except one, clusters out of these 20 clusters, showed a statistically significant positive amplitude- $\rho^2_{adj}$ correlation. Cluster frequency for an oscillation cluster was defined to be the average of the oscillation frequencies across all electrodes for the given oscillation cluster. (b). Wave-strength (denoted as $\rho^2_{adj}$) of high (beta) frequency traveling waves is uncorrelated with the wave-strength ($\rho^2_{adj}$ (low)- $\rho^2_{adj}$ (high)), phase (phase (low)- $\rho^2_{adj}$ (high)), and power (power (low)- $\rho^2_{adj}$ (high)) of low (theta) frequency traveling waves. For testing the statistical significance of the observed correlations, values from one time-series were circularly shuffled with respect to the other so that the instantaneous correlation between the two time-series is destroyed, and correlation analysis was repeated on this shuffled data to build a distribution of surrogate correlation values against which the observed correlation was tested (p<0.05) (Materials and methods).

• Transparent reporting form

## Data availability

Deidentified data are fully available to the public without any restrictions and can be downloaded here https://dabi.loni.usc.edu/dsi/anon?token=Do2yMlnZiXwKwFmOeCDtK. Codes used for the analyses are also fully available to the public without any restrictions and can be downloaded here https://github.com/john-myers-github/INSULA_RS, (copy archived at swh:1:rev:92bd74375262afe7ff0882d1ff151f774fcc0885).

The following dataset was generated:

| Author(s) | Year | Dataset title | Dataset URL | Database and Identifier |
|---|---|---|---|---|
| Myers J | 2022 | Spontaneous neuronal oscillations in the human insula are hierarchically organized traveling waves | https://github.com/john-myers-github/INSULA_RS | GitHub, INSULA_RS |

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
