## [Editor Report]

This manuscript is of interest to neuroscientists willing to deepen their knowledge related to the role of the insula and to any scientist interested in oscillatory activity and traveling waves in the brain. The substantial dataset and the novel methodological approach provide interesting insights on the functional organization of this brain region.

---

## [Decision Letter]

**Decision letter after peer review:**

Thank you for submitting your article "Spontaneous Neuronal Oscillations in the Human Insula are Hierarchically Organized Traveling Waves" for consideration by *eLife*. Your article has been reviewed by 2 peer reviewers, and the evaluation has been overseen by a Reviewing Editor and Laura Colgin as the Senior Editor. The following individual involved in the review of your submission has agreed to reveal their identity: Anais LLorens (Reviewer #2).

Essential revisions:

1) The reviewers raise important points about the results being driven by particular subjects or electrodes. This is particularly true for the traveling waves, which are sparse, but also applies to other analyses. Any revision needs to address this issue.

2) It is unclear why the 9-15 Hz frequency range is currently excluded. This range should either be included, or its exclusion should be justified and any resulting bias in the algorithm addressed.

3) The possibility that power differences are driven by the relative positions of the electrodes should be investigated.

*Reviewer #1 (Recommendations for the authors):*

The manuscript should provide figures of the direction of the traveling waves across all subjects, not only in participants 3, 6, 7.

The z-axes in Figure 2 differ, a consistent presentation would help.

Since there are no SI power gradients, the data could additionally be collapsed across the SI and shown only for the AP gradient to more clearly show the results in addition to the 3D plots.

It is unclear in the methods whether electrode positions are assessed using the MNI152 conversion or using the more detailed Freesurfer extraction.

The connection between the first set of studies of gradient analyses and the second set of traveling wave studies is not clear. The preprocessing steps and referencing scheme are different for each set. It seems like the reader should think that prominent oscillations are first identified and then it is studied specifically whether these oscillations present as traveling waves.

*Reviewer #2 (Recommendations for the authors):*

Overall, this manuscript is interesting, well written and well executed. I am impressed by the insular coverage and the methodological approach is novel and well explained and allows to answer most of the questions asked.

In the following sections I will detail the couple of remarks and questions I have regarding ways to improve the manuscript. My general comment concerns the lack of information regarding the localization of some effects. Indeed, the first section of the manuscript gives many details regarding the laterality of the effect as well as on the y- and z-axes, while the second section does not. Albeit being analyzed at the individual level with contacts that can be included in several clusters, I think that more information regarding the anatomical location of the clusters and the traveling waves need to be reported.

1 – The Coverage:

The overall coverage figure 1 shows some electrodes that seem to be in the temporal or frontal opercula, I know that it might be due to the projection onto the MNI brain but specifying the percentage of electrodes in the different subregions of the insula (or at least in the anterior and posterior parts of it) would still be informative.

2– Clusters and traveling waves:

– The manuscript could benefit from some precisions regarding the localization of the clusters. For instance, when the authors report having found 28 oscillation clusters across the 10 participants, they should report the proportion by hemisphere and, if possible, on the anteroposterior axis. Same comment for the 35% of clusters presenting traveling waves.

– In my opinion, adding a histogram of the instantaneous direction of propagation of the traveling waves across time for all the clusters across all the participants, for one hemisphere at the time, could clarify visually the lack of convergence of direction between clusters.

– I also wonder whether using a smaller number of connected electrodes (3 instead of 4 for instance) could improve the anatomical resolution of the clusters.

---

## [Author Response]

Essential revisions:1) The reviewers raise important points about the results being driven by particular subjects or electrodes. This is particularly true for the traveling waves, which are sparse, but also applies to other analyses. Any revision needs to address this issue.

We thank the reviewers for stating this concern and we have revised our paper to explain how we verified that our results are robustly present across electrodes in the population. We would like to clarify here, and as mentioned in the original version of the manuscript (please see *Traveling waves analysis* section in Methods), the traveling waves analysis was carried out on each subject individually (Figures 4-7, S1-S9; Tables S1-S2). Because traveling waves were found in 9 of 9 subjects, it precludes the possibility that the results showing the presence of traveling waves are driven by any individual subject. Moreover, as mentioned in the Statistical analysis section, we used a surrogate analysis to test the statistical significance of the detected traveling waves for each individual cluster in each subject. The coordinates of the electrodes with respect to the Hilbert-transformed phases were shuffled so that the contiguous spatial variation of the phases is destroyed. The circular-linear regression analysis was repeated on this shuffled data to build a distribution of surrogate wave-strength values against which the observed wave-strength was tested (*p* < 0.05). Importantly, this shuffling procedure ensured that our statistical test specifically evaluated the spatial arrangement of the recording electrodes and that the results precluded the possibility that the results are driven by a specific subset of electrodes or their spatial arrangement. The text in manuscript has now been modified to reflect these clarifications (page 8), by more clearly describing how our statistical procedure confirms that traveling waves are robustly present across subjects and electrodes.

The revised paper now more clearly describes our methods for analyzing frequency and power gradients, explaining how our methods ensure robustness. We used a linear mixed-effect model to analyze these patterns, which included a categorical factor for subject identity. This factor ensured that our results were not caused by individual subjects. Therefore, as we now explain more clearly in the revised paper (pages 6-7), consistent with our main claims, all reported t-statistics specifically reflect the overall strength (effect size) and spatial direction of spectral gradients across electrodes, while accounting for differences between individual subjects. The number of electrodes contributed by each subject was also similar (mean = 23.9 contacts, standard error = +/- 2.43 contacts), which suggests our results were not driven by any one subject. Finally, to ensure that we address this concern fully we added a supplementary figure (Figure S11) showing that single-subject gradients directionally align with the group level results. The text in manuscript has now been modified to reflect these changes (pages 6-7).

2) It is unclear why the 9-15 Hz frequency range is currently excluded. This range should either be included, or its exclusion should be justified and any resulting bias in the algorithm addressed.

We thank the reviewers for this concern. We would like to clarify that our analysis does not contain any inherent frequency band bias. This 9-15 Hz band was not excluded, but instead we examined it and did not observe significant oscillation clusters at frequencies in this range. We now clarify this in the revised text (pages 5, 9, 15). The absence of insular oscillations in this band can be seen in Figures S1-S9 and Tables S1-S2, where oscillations were mostly present in the theta and β bands, but not in 9-15 Hz.

Actually, we are thankful to the reviewer for emphasizing this point to us. We think it is notable that there were no 9-15 Hz oscillations in the insula, and we modified the discussion to discuss this point (page 9).

3) The possibility that power differences are driven by the relative positions of the electrodes should be investigated.

We thank the reviewer for this interesting point and we conducted new analyses of the impact of white matter on our results. Specifically, we have added more detail to the subsection ‘Electrode localization’ in the Methods section to describe the proportion of white matter contacts in our data. The revised text reads:

“An expert rater (B. S.) examined the fused MRI to CT images and determined whether each contact was in white or gray matter based on its location on the brain slice. Contacts were labeled as 'gray matter' if they were located within 3 mm of cortical gray matter. Contacts labeled 'white matter' had no voxels within 3 mm of grey matter.”

As we now report in the paper (page 15), “the vast majority (97%) of all contacts in this study were in insula gray matter.” Therefore, it is unlikely that the power differences are driven by the relative positions of the electrodes with respect to the white matter.

Reviewer #1 (Recommendations for the authors):The manuscript should provide figures of the direction of the traveling waves across all subjects, not only in participants 3, 6, 7.

As requested, the revised manuscript now includes histogram of the instantaneous propagation directions of the traveling waves across time, clusters, and participants, for each hemisphere separately (Figures 6k-l).

The z-axes in Figure 2 differ, a consistent presentation would help.

We have now modified the z-axes in Figure 2 for consistency.

Since there are no SI power gradients, the data could additionally be collapsed across the SI and shown only for the AP gradient to more clearly show the results in addition to the 3D plots.

In the right insula, we did observe an SI gradient for theta power (Figure 2D). The geometric planes help present the neural data alongside the spatial information in a way that we feel is maximally informative. In addition to the orientations of the planes, the color axes also provide information on how the neural data varies across space.

Although we appreciate the reviewer’s suggestion, after consideration, we feel that collapsing across dimensions only for significant effects may confuse some readers as they switch between 2D and 3D representations of the same data. Therefore, unless there is something missing or the reviewer feels strongly, we respectfully would prefer to keep the plots in their current forms.

It is unclear in the methods whether electrode positions are assessed using the MNI152 conversion or using the more detailed Freesurfer extraction.

We have edited the text of the methods on page 15 to more directly clarify that all neuroimages were coregistered to standard MNI 152 space in an iterative semi-automated manner. Additionally, to verify the results of our co-registration approach, electrodes located in the insula were first visually identified by a neurosurgeon and further verified using the latest Desikan-Killiany (DK) atlas, available in the Freesurfer software (Desikan et al., 2006; Fischl, 2012).

The connection between the first set of studies of gradient analyses and the second set of traveling wave studies is not clear. The preprocessing steps and referencing scheme are different for each set. It seems like the reader should think that prominent oscillations are first identified and then it is studied specifically whether these oscillations present as traveling waves.

We thank the reviewer for this concern. As suggested, we have revised the paper to more clearly transition between the two sections of the paper and explain how the first set of studies on power and frequency gradients inspired the second set on traveling waves. Here we now cite work by Ermentrout and Kopell (1984) which suggested that these phenomena are linked. These topics are discussed in the revised Results as well as the Discussion (pages 7, 12).

As the reviewer has correctly pointed out, we first identify the oscillations of the electrodes across subjects to study the spatiotemporal gradients of frequency and power and next, further examine whether these oscillations also exhibit traveling waves in each participant individually. Lack of sufficient electrode placements (Figures S1-S9) simultaneously in both the anterior–posterior and dorsal–ventral portions of the insula in individual participants prevented us from directly examining the connection between the analyses of spatiotemporal gradients and traveling waves. Future studies with adequate sampling of electrodes in the insula of individual participants are needed to rigorously analyze the instantaneous propagation directions of traveling waves and relate them directly to the frequency and power gradients that we observed in the insula.

Reviewer #2 (Recommendations for the authors):Overall, this manuscript is interesting, well written and well executed. I am impressed by the insular coverage and the methodological approach is novel and well explained and allows to answer most of the questions asked.In the following sections I will detail the couple of remarks and questions I have regarding ways to improve the manuscript. My general comment concerns the lack of information regarding the localization of some effects. Indeed, the first section of the manuscript gives many details regarding the laterality of the effect as well as on the y- and z-axes, while the second section does not. Albeit being analyzed at the individual level with contacts that can be included in several clusters, I think that more information regarding the anatomical location of the clusters and the traveling waves need to be reported.

We thank the reviewer for the encouraging comments. Below, please find point-by-point responses to the concerns.

1 – The Coverage:The overall coverage figure 1 shows some electrodes that seem to be in the temporal or frontal opercula, I know that it might be due to the projection onto the MNI brain but specifying the percentage of electrodes in the different subregions of the insula (or at least in the anterior and posterior parts of it) would still be informative.

We thank the reviewer for this concern. As the reviewer has correctly pointed out, Figure 1 could be misleading due to the projection of individual patient space to the MNI 152 space. Therefore, we added a new supplementary figure (Figure S10) showing the anatomical distribution of electrodes across the insula. This figure shows wide coverage of electrodes across both the anterior (which includes the shortanterior, short-middle, and short-posterior gyri subdivisions) and posterior (which includes the longanterior and long-posterior subdivisions) regions as well as the apex region of the insula. We have now edited the text of the methods on page 14 to reflect these changes.

2– Clusters and traveling waves:– The manuscript could benefit from some precisions regarding the localization of the clusters. For instance, when the authors report having found 28 oscillation clusters across the 10 participants, they should report the proportion by hemisphere and, if possible, on the anteroposterior axis. Same comment for the 35% of clusters presenting traveling waves.

We have revised the manuscript to provide new data to clarify this concern. The proportions of oscillation clusters and traveling waves by hemisphere have now been included in pages 7-8 in the revised manuscript. We would also like to clarify that the clustering analysis in our study is done by calculating the proximity of the electrodes with respect to frequency and distance, and not by their position along the anteroposterior axis. Consequently, we do not distinguish electrodes or clusters based on their position in anterior or posterior insula for the traveling waves analysis. We have now edited the text in the paper on page 16 to clarify these points.

– In my opinion, adding a histogram of the instantaneous direction of propagation of the traveling waves across time for all the clusters across all the participants, for one hemisphere at the time, could clarify visually the lack of convergence of direction between clusters.

Thank you for this idea. As you suggested, a histogram of the instantaneous propagation directions of the traveling waves across time, clusters, and participants, for each hemisphere separately, has now been included in the revised version of our manuscript (Figures 6k-l).

– I also wonder whether using a smaller number of connected electrodes (3 instead of 4 for instance) could improve the anatomical resolution of the clusters.

We appreciate this suggestion, as this is in fact something we experimented with previously. However, empirically, we have observed that there is a greater chance of observing false positive traveling waves if we consider a relatively low number of connected electrodes such as 3. Hence, we have kept the number of connected electrodes 4 in our analysis, as we found that this provides a good trade-off between anatomical resolution and the false positive rate for measuring traveling waves.